# Coordination of shoot apical meristem shape and identity by APETALA2 during floral transition in Arabidopsis

Enric Bertran Garcia de Olalla [1,2,4], Martina Cerise [1,4],
Gabriel Rodríguez-Maroto [1], Pau Casanova-Ferrer [1], Alice Vayssières[1],
Edouard Severing[1,3], Yaiza López Sampere[1], Kang Wang[1], Sabine Schäfer[1],
Pau Formosa-Jordan [1] & George Coupland [1] ✉

Plants flower in response to environmental signals. These signals change the shape and developmental identity of the shoot apical meristem (SAM), causing it to form flowers and inflorescences. We show that the increases in SAM width and height during floral transition correlate with changes in size of the central zone (CZ), defined by *CLAVATA3* expression, and involve a transient increase in the height of the organizing center (OC), defined by *WUSCHEL* expression. The APETALA2 (AP2) transcription factor is required for the rapid increases in SAM height and width, by maintaining the width of the OC and increasing the height and width of the CZ. AP2 expression is repressed in the SAM at the end of floral transition, and extending the duration of its expression increases SAM width. Transcriptional repression by SUPPRESSOR OF OVEREXPRESSION OF CONSTANS1 (SOC1) represents one of the mechanisms reducing AP2 expression during floral transition. Moreover, AP2 represses *SOC1* transcription, and we find that reciprocal repression of SOC1 and AP2 contributes to synchronizing precise changes in meristem shape with floral transition.

The shoot apical meristem (SAM) contains a population of stem cells that gives rise to all above-ground tissues. As cells are displaced from the stem cell niche, they differentiate and form organs on the flanks of the SAM, but the stem cell population is maintained throughout the life of the plant to allow continuous organ production[1,2]. The structure and function of the SAM change with plant age and in response to environmental signals[3,4]. Notably, during the transition to flowering, the SAM enlarges and changes identity to initiate the formation of flowers instead of leaves[5–7]. The increase in SAM size during floral transition persists in the inflorescence meristem, and likely contributes to the number of flowers formed[5,8–10]. However, how SAM shape changes as it increases in size, how these changes impact on different meristematic domains and how

changes in meristem shape and identity are temporally coordinated remain unclear.

Exposure of Arabidopsis plants to long days (LDs) causes the SAM to transition rapidly from the vegetative to the reproductive state, which involves a radical reprogramming of the SAM transcriptome[11,12]. During this process, genes encoding transcription factors that repress flowering are downregulated, whereas the expression of floral promoters is upregulated. One of the earliest induced genes is *SUPPRESSOR OF OVEREXPRESSION OF CONSTANS1* (*SOC1*), which encodes a MADS-domain transcription factor and promotes floral transition at the SAM[13–15]. During floral transition, as well as increasing in size, the SAM forms a characteristic dome shape[5,16–19]. Mutations in *SOC1* delay the increase in SAM area during floral transition, and alter the

[1]Department of Plant Developmental Biology, Max Planck Institute for Plant Breeding Research, Cologne, Germany. [2]Present address: Laboratoire Reproduction et Développement des Plantes, Univ Lyon, ENS de Lyon, CNRS, INRAE, INRIA, Lyon, France. [3]Present address: Department of Plant Breeding, Wageningen University and Research, Droevendaalsesteeg 4, PB Wageningen, The Netherlands. [4]These authors contributed equally: Enric Bertran Garcia de Olalla, Martina Cerise. ✉e-mail: coupland@mpipz.mpg.de

expression levels of many flowering genes[20,21] as well as enzymes involved in gibberellin biosynthesis and catabolism[5].

Although several mechanisms that influence SAM size have been described[22–31], those responsible for altering SAM morphology during floral transition remain to be elucidated. In the inflorescence SAM, the feedback loop between the WUSCHEL (WUS) homeodomain transcription factor and the CLAVATA3 peptide (CLV3), which are expressed in the organizing center (OC) and the central zone (CZ) respectively, maintains the size of the SAM[22,32–35]. *WUS* expression can be increased by the APETALA2 (AP2) transcription factor[23,36–38], which is expressed in the SAM during vegetative development but is absent from the inflorescence SAM[8,39,40]. Moreover, *ap2* mutants are impaired in sepal and petal identity[41,42], are early flowering and have smaller meristems at the embryonic and inflorescence stages[8,23,43]. By contrast, gain-of-function *AP2* transgenes that are resistant to microRNA172 (miR172) confer an increase in inflorescence meristem size and late flowering[8,23].

Here, we quantify the changes in shape of the Arabidopsis SAM during floral transition and show that AP2 is required to increase the size of the CZ and maintain the width of the OC. Moreover, we find that AP2 and SOC1 mutually repress each other's expression in the SAM during floral transition and that this coordinates changes in SAM shape with the switch in primordium identity.

## Results

### AP2 is required for the rapid increases in meristem height and width that occur during floral transition

The Col-0 SAM increases in area during floral transition, but the concomitant changes in SAM shape have not been assessed[5,16–19]. To quantify changes in SAM shape, we measured the height and width of Col-0 SAMs harvested in a time series under long-day conditions (LD) (Fig. 1, Methods). Both SAM height and width increased progressively and disproportionally from 10 to 14 LD after germination, when they both reached their maximum values (Figs. 1a, b; Supplementary Figs. 1a, b). Therefore, SAM width increased almost 1.5-fold from 10 to 14 LD, whereas SAM height increased on average by 2-fold (Figs. 1a, b; Supplementary Fig. 1b). These changes coincide with the increase in SAM area (Fig. 1d, e) and resulted in a domed SAM with maximum height at 14 LD that could be represented as a parabola[44,45] (Methods; Fig. 1c). Similar results were obtained after transferring 2-week-old Col-0 plants from short-day conditions (SD) to LD (Supplementary Fig. 2). Under these conditions, the SAM remained vegetative until transfer to LD, and reached maximum height, width and area after exposure to 7 LD (Supplementary Figs. 2b, e, f).

The inflorescence SAM of the *ap2-12* mutant is smaller than that of Col-0[8,23], but whether this difference arose during floral transition and its effect on SAM shape have not been examined. Therefore, the same approaches as described above were used to analyze the *ap2-12* SAM during floral transition (Fig. 1, Supplementary Fig. 1). The width and area of the SAM of *ap2-12* mutants were not significantly different to Col-0 at 12 LD, whereas the height of the *ap2-12* SAM was slightly less. However, at 14 LD, when Col-0 SAM increased markedly in height, width and area, the SAM of *ap2-12* mutants remained similar to 12 LD (Figs. 1a–b, e). At 14 LD, the *ap2-12* SAM was approximately 50% and 25% smaller in height and width than the Col-0 SAM, respectively. Similarly, no significant difference between the SAM shape of *ap2-12* mutants and Col-0 was observed after vegetative growth for 2 wSD (Supplementary Fig. 2), but at +7 LD, when the Col-0 SAM reached maximum height, the *ap2-12* SAM showed a 50% reduction in both height and area compared to Col-0 and did not show the characteristic domed shape of the Col-0 SAM (Supplementary Fig. 2). These results demonstrate that AP2 is required for the large increases in height, width and area of the Col-0 SAM that occur during floral transition.

AP2 levels are negatively regulated by miR172 in the inflorescence SAM[40]. To test whether preventing the repression of AP2 by miR172 altered SAM shape during floral transition, apices from plants carrying a miR172-resistant version of *AP2* fused to VENUS and expressed from the *AP2* promoter (*rAP2-V*)[8] were analyzed (Figs. 1a–c, Supplementary Fig. 1a, d). At 10 LD, the SAM of these plants was similar in width to the Col-0 SAM, but it was larger than Col-0 at all time points from 12 LD to 19 LD (Fig. 1a). Moreover, from 17 LD, the height and area of the *rAP2-V* SAM were also larger than that of Col-0 (Fig. 1b). Similarly, after transfer to LDs, *rAP2-V* SAM width was higher than Col-0 prior to (+ 5 LD) and after (+ 9LD, + 11 LD) floral transition, and height was larger at +9 LD (Supplementary Figs. 2b, e, f). The results indicate that preventing repression of AP2 by miR172 increases SAM width during floral transition, and that after floral transition of Col-0, *rAP2-V* maintains the SAM at a larger size with greater height and width, consistent with the previous report that the inflorescence SAM of *rAP2-V* is larger than that of Col-0[8].

In Col-0, enlargement of the SAM during floral transition is associated with increases in both number and area of cells in the epidermis (Layer 1 (L1); Figs. 1d–f, Supplementary Figs. 1e, Supplementary Fig. 2a–d)[5]. The SAM L1 of *ap2-12* contained significantly fewer cells than that of Col-0 during floral transition under LD and after transfer from SD to LD (Fig. 1f, and Supplementary Fig. 2c). Moreover, in most of the time points, cell area was significantly increased in *ap2-12* and *rAP2-V* compared with Col-0 (Supplementary Fig. 1e, and Supplementary Fig. 2d). Therefore, AP2 regulates both cell number and cell size in the L1 during floral transition.

In summary, AP2 is required in Col-0 to rapidly increase SAM width and height during floral transition and to form a characteristically domed SAM, whereas increasing the amount and persistence of AP2 due to insensitivity to miR172 most strongly increases SAM width compared to Col-0 prior to floral transition, and maintains SAM width, height and area after floral transition of Col-0.

### AP2 is present in the SAM as it increases in height and width during floral transition

AP2 was previously detected in the vegetative SAM prior to floral transition, and found to be absent from the mature inflorescence SAM[8,40], but the dynamics of its reduction during floral transition and how this relates to the increases in SAM height and width were not reported. *AP2::AP2:VENUS* expression was examined in the SAM of LD-grown plants from 7 until 19 LD, and the fluorescence intensity of AP2-VENUS was quantified at each time point using a computational pipeline that we designed to assess the mean fluorescence levels at the tip of the SAM (Methods; Figs. 2a–c). AP2-VENUS abundance reduced progressively from 7 LD until 17 LD, and was still detected at 14 LD when the SAM of Col-0 is at maximum height. Moreover, after transfer from SD to LD, AP2-VENUS was present at similar levels after 3 and 5 LD, but then sharply reduced in abundance at 7 LD (Supplementary Fig. 3). By contrast, the rAP2-VENUS protein persisted in the SAM until the end of the time course at +13 LD. These results demonstrate that although AP2 protein represses flowering, it is still present during floral transition when the SAM increases rapidly in height and width, and declines around the stage that the SAM reaches maximum height. Together with the reduced height and width of the SAM of *ap2-12* mutants (Figs. 1a, b, Supplementary Figs. 2e–f), these results suggest that AP2 acts in the SAM during floral transition to confer the characteristic domed SAM shape observed in Col-0 (Fig. 1c, Supplementary Fig. 2g). Moreover, the persistence of rAP2-VENUS after floral transition of Col-0 (Supplementary Fig. 3)[8] and the increased SAM height and width of *rAP2-V* (Figs. 1a, b; Supplementary Figs. 2e, f) at this stage suggests that reduction of AP2 levels after floral transition is required to terminate the rapid lateral and vertical growth of the Col-0 SAM that occurs during floral transition.

### Relationship of organ primordia identity to meristem shape during floral transition

To assess how the developmental stage of the SAM correlated with changes in SAM shape, the identity of organ primordia present on the

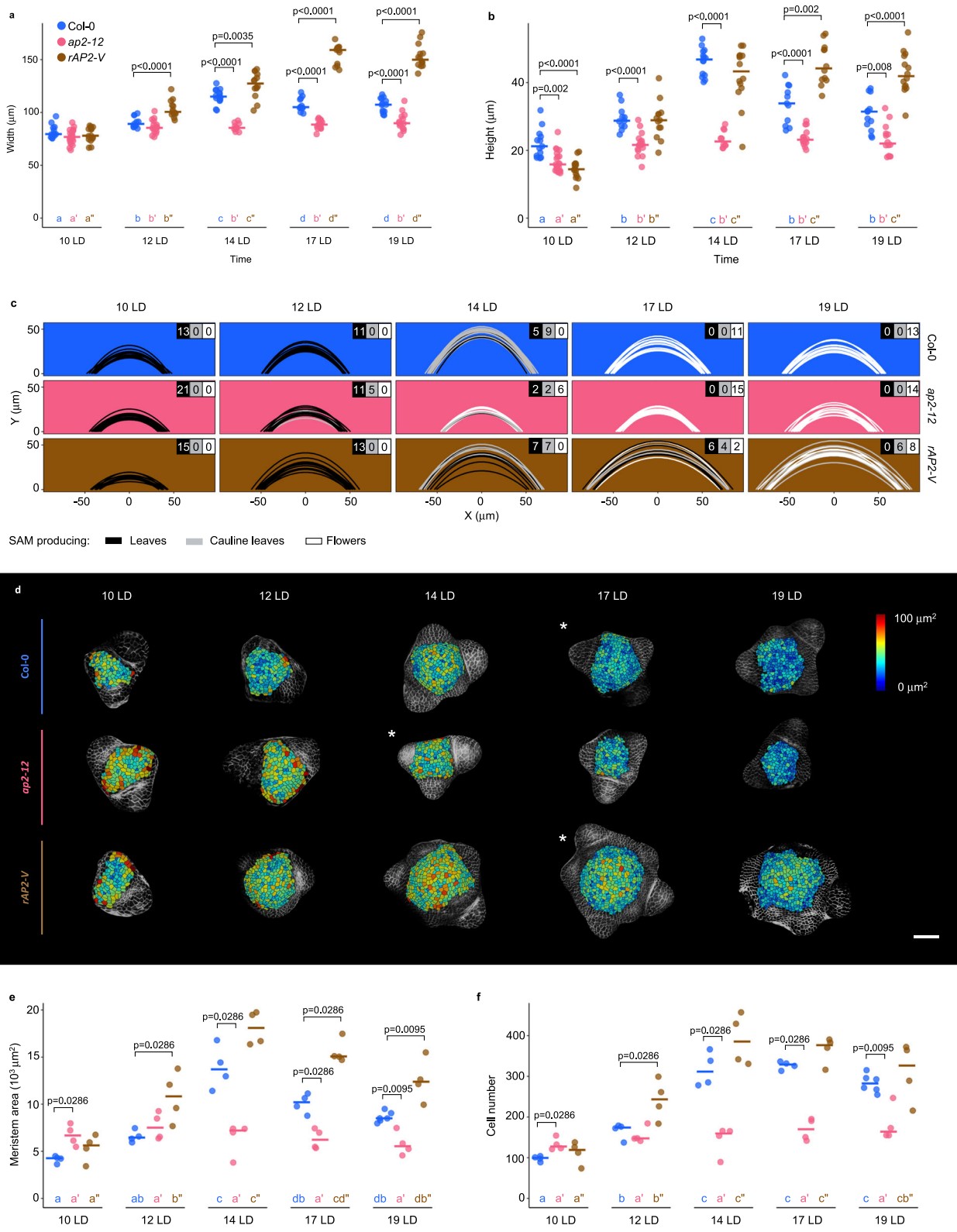

flanks of the SAM was examined throughout the LD time course. Each SAM was scored microscopically for whether vegetative leaves, cauline leaves or floral primordia were visible (Supplementary Fig. 4). At 14 LD, when the Col-0 SAM reached maximum average height, all SAMs formed visible vegetative or cauline leaves, and SAMs forming exclusively floral primordia were only present from 17 LD (Fig. 1c). More Col-0 SAMs at 14 LD were analyzed by combining individuals from several

independent experiments (Fig. 2d). Again, the majority of Col-0 SAMs at the time of maximum average height had visible leaves or cauline leaves (around 90%), and a smaller proportion (10%) had exclusively floral primordia. The SAMs of those individuals with only visible floral primordia were generally smaller than those with visible leaf or cauline leaf primordia (Mann-Whitney-Wilcoxon test, $p = 0.0024$), suggesting that the SAM size of individuals with only floral primordia had already

**Fig. 1 | AP2 is a positive regulator of SAM size and morphology during floral transition. a–b** Measurement of (**a**) width and (**b**) height of the SAM in continuous LD-grown plants. **c** SAM morphology adjusted to parabolas. The parabolas are colored according to the identity of primordia that were formed at the SAM periphery. The number of meristems producing each kind of primordia are listed on the top-right corner in each genotype and time point. **d–f** Segmentation analysis of the SAM of Col-0, *ap2-12* and *rAP2-VENUS* under continuous long days (LDs). *n* = 4 SAMs. **d** Top view of the heatmap quantification of cell area in the meristem region. White asterisks indicate the first time point at which floral primordia were detected in the analysis of the corresponding genotype. Scale bar = 50 μm. **e–f** Quantification

of (**e**) meristem area and (**f**) cell number. **a–b, e–f** The horizontal bars represent the median value for each genotype. Significant differences between wild-type and mutants within each time point were determined via two-sided Mann-Whitney-Wilcoxon-test ($p < 0.05$). Significant differences among time points within each genotype were determined via one-way ANOVA (two-sided), followed by Tukey post-hoc comparisons ($p < 0.05$). Data sets that share a common letter do not differ significantly. The color of the dots and the letters correspond to the genotype. See Supplementary Data 7 for precise sample size and *p*-values of Mann-Whitney-Wilcoxon and ANOVA test. Source Data are provided as a Source Data file.

passed maximum height. Therefore, most or all Col-0 SAMs undergoing floral transition reach maximum height while leaf or cauline leaf primordia are visible at the apex.

A similar analysis was performed for *ap2-12* mutants. *AP2* is a negative regulator of flowering, and *ap2* mutants flower earlier than Col-0[43]. The *ap2-12* mutant SAM increased in height to a much lesser extent than that of Col-0, but reached maximum height at 12 LD (Fig. 1b). In the *ap2-12* mutant, the formation of floral primordia and maximum SAM height both occurred around 2–3 days earlier than in Col-0, but the SAM did not reach the height of the Col-0 SAM. These observations raised the possibility that continued increase in SAM height in *ap2-12* mutants might be prevented by the SAM transitioning more rapidly to forming floral primordia. To test this possibility, the shape of the SAM of *short vegetative phase-41* (*svp-41*), another early-flowering mutant[46], was examined during floral transition. Only floral primordia were detected at the SAM of most (87.5%) *svp-41* mutants at 12 LD and at all *svp-41* SAMs at 14 LD (Supplementary Fig. 5a). These mutants therefore form floral primordia 3–5 days earlier than Col-0, as *ap2-12*. However, *svp-41* SAMs increase strongly in height and width during floral transition, reaching a maximum height at 10 LD that is comparable to that of Col-0 at 14 LD (Supplementary Figs. 5b, c). Therefore, early transition to forming floral primordia does not prevent an increase in SAM height in *svp-41* mutants. The comparison between *svp-41* and *ap2-12* suggests that AP2 has two distinct functions during floral transition, one to promote SAM height and width, and another to repress the transition to forming floral primordia. Loss of AP2 function allows flowering to proceed without formation of a fully domed SAM (Fig. 1c, Supplementary Fig. 2g), effectively uncoupling the sequential progression of SAM doming and formation of floral primordia observed in Col-0.

## AP2 is required for increases in central zone height and width and peripheral zone width during floral transition

The SAM changes in morphology during floral transition, but how this affects its internal organization has not been described in detail. In different genetic backgrounds and in response to nutrient availability or light quality, the size and morphology of the inflorescence SAM correlates with the shape and size of the CZ and OC[3,4,44]. To test how the CZ and OC change in size and shape during floral transition, the transcriptional fusions *CLV3::mCHERRY:NLS* and *WUS::3xVENUS:NLS*[4] were used as markers for the CZ and OC, respectively (Fig. 3; Supplementary Fig. 6). A quantitative pipeline was then developed to assess the size of these fluorescence domains (Methods). In the Col-0 SAM, the CZ increased in both height and width from 10 LD to 16 LD as floral transition progressed and SAM height and width increased (Figs. 3b, c; Supplementary Fig. 6a). After floral transition, the final height and width of the CZ were larger in the mature inflorescence SAM at 16 and 20 LD than in the vegetative SAM. The OC of the Col-0 SAM also showed dynamic changes in shape during floral transition. A transient increase in OC height occurred at 13 LD, as the SAM approached its maximum height, but no difference in OC width was observed at any time point, although the SAM increased in width between 10 LD and 16 LD (Figs. 3a, d, e; Supplementary Figs. 6b–e). The discrepancy between the increase in the width of the SAM and of the OC suggested that the

width of the peripheral zone (PZ) increased during floral transition. A quantification of the peripheral width (Methods) demonstrated that the PZ increased during floral transition from 10 LD to 16 LD (Fig. 3f, and Supplementary Fig. 6b). Moreover, because the OC width stayed broadly similar and the PZ width increased, the ratio of PZ to OC increased from 10 LD to 16 LD (Fig. 3g). Also, in the mature inflorescence SAM at 20 LD the width and height of the OC were similar to those of the vegetative SAM at 10 LD, but the width of the PZ was greater (Figs. 3d–f). These data suggest that during floral transition, the CZ increases in height and width and the PZ increases in width, and these changes persist into the inflorescence SAM. Yet, the OC does not increase in width and only temporarily increases in height when the SAM is growing most rapidly vertically, and recedes before the mature inflorescence SAM is formed.

The effects of *ap2-12* on the CZ, OC and PZ were then examined. In the SAM of *ap2-12*, the CZ is significantly larger than that of Col-0 at 10 LD and did not increase much in height across the time course (Figs. 3b, c, Supplementary Fig. 6a). At 16 LD, when AP2 protein levels are being strongly reduced in the SAM of wild type (Figs. 2a–c), the CZ height was approximately 15% smaller than that of Col-0. Moreover, the width of the CZ of *ap2-12* increased to a lesser extent than that of Col-0 and was approximately 20% smaller than Col-0 at 16 LD (Fig. 3c). At 20 LD, when AP2 expression is expected to be entirely repressed, the height and width of the *ap2-12* CZ was similar to those of the 10 LD SAM, in contrast to the Col-0 SAM, in which these parameters had increased by 38% and 29%, respectively (Figs. 3b, c, Supplementary Fig. 6a). The width of the PZ was also significantly smaller at 16 LD and 20 LD in *ap2-12* compared to Col-0, and again did not increase across the time course, in contrast to Col-0 (Fig. 3f). Moreover, although the OC width was also 7% smaller in *ap2-12* than Col-0 at 16 LD, the ratio of PZ to OC was reduced at 16 LD compared to Col-0 (Fig. 3g), suggesting that the width of the PZ is reduced disproportionately compared to the OC. The height of the OC was 40% greater than that of Col-0 at 10 LDs and gradually decreased during the time course, suggesting that the earlier flowering of *ap2-12* accelerated the increase in OC height observed in Col-0 at 13 LD (Fig. 3d). By 20 LD, all SAM parameters except OC height were smaller in *ap2-12* than in Col-0 (Figs. 3b–f, Supplementary Figs. 6c–d), but, nevertheless, the balance among different meristematic regions appeared to be restored (e.g., the ratio of PZ to OC was the same for Col-0 and *ap2-12* at 20 days) (Fig. 3g). These data indicate that the largest reductions between *ap2-12* and Col-0 SAMs are observed in CZ height and width, as well as PZ width in the early inflorescence SAMs at 16 LD, but that by 20 LD all meristematic zones are smaller in *ap2-12* than Col-0.

The effect of *rAP2* on SAM organization was then tested. At 13 LD and 16 LD when AP2 levels are falling in wild type (Fig. 2a, b), the width of meristematic domains is significantly increased in rAP2 SAMs (Figs. 3c, e) without greatly affecting their height. Only the CZ height shows a gradual increase, reaching an increase of 17% at mature inflorescence stage (20 LD). Already at 13 LD, the width of the CZ and the OC are greater than in Col-0, and at 16 LD both the PZ and the OC show a larger increase in width, 20% and 25% respectively (Figs. 3c, e, f). The mature inflorescence SAM of *rAP2* at 20 LD is much wider than the vegetative SAM in the OC, CZ and PZ

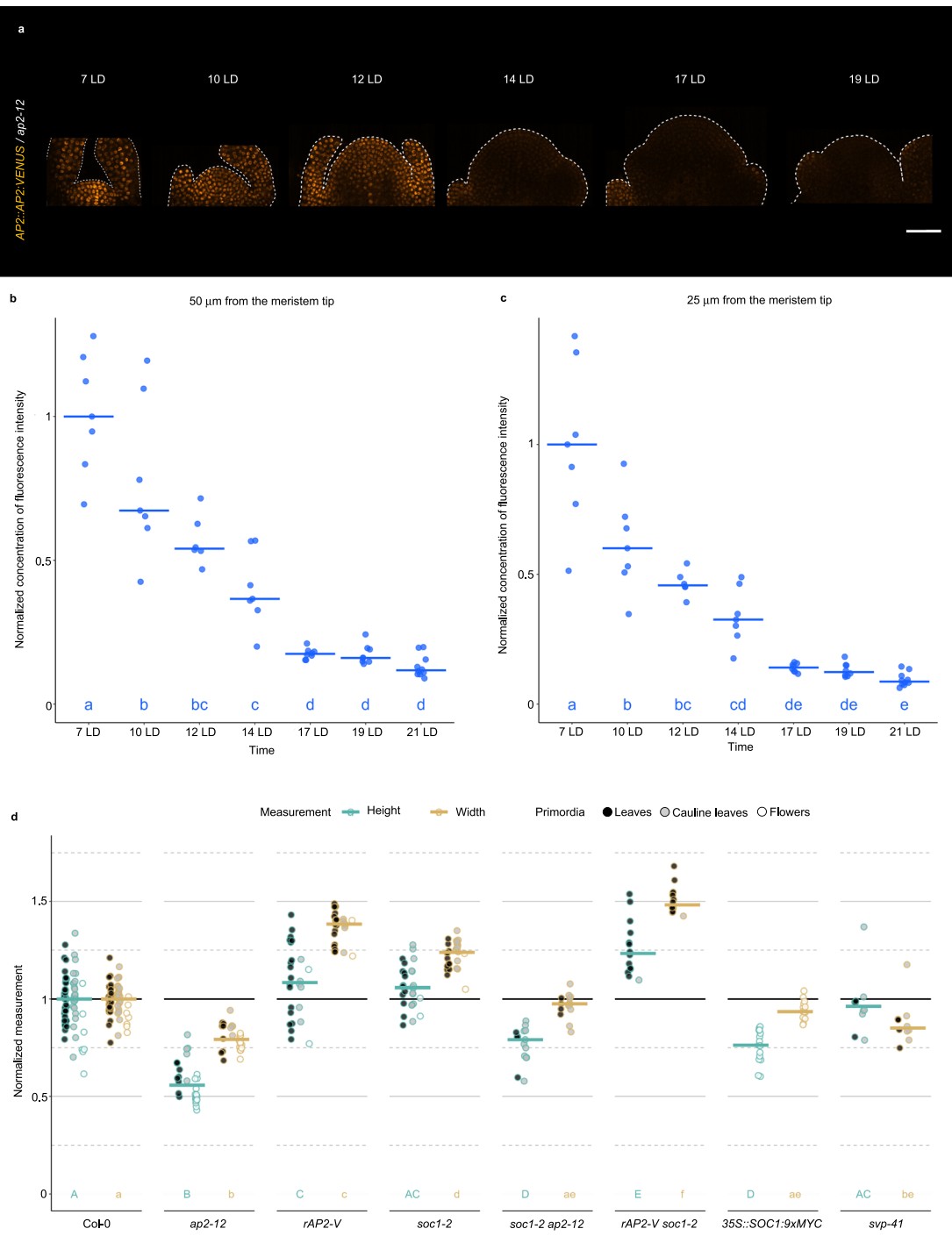

**Fig. 2 | AP2 is present in the SAM as it increases in height and width during floral transition. a** Pattern of protein accumulation of *AP2::AP2:VENUS* at the SAM in *ap2-12* plants under continuous LDs. Each SAM is shown in longitudinal section. The outline of each acquired meristem and its peripheral organs is indicated with a dotted white line. Scale bar = 50 μm. **b**–**c** Quantification of AP2:VENUS concentration of fluorescence intensity (total fluorescence divided by volume) at the shoot apex from the tip (**b**) to 50 μm deep or (**c**) 25 μm deep in the basal direction in the *ap2-12* mutant background during continuous LDs. The horizontal bars represent the median value for each time point. Significant differences among time points within each genotype were determined via one-way ANOVA (two-sided), followed by Tukey post-hoc comparisons (*p* < 0.05). Data sets that share a common letter do not differ significantly. **d** Normalized measurements of height

and width of the SAMs from LD-grown plants in this study (data from 6 independent experiments were pooled). The point of maximum height of each genotype in each experiment was selected. Measurements were normalized by the median of that measurement in Col-0 in each experiment. The horizontal bars represent the median value for each genotype. The outlines of the dots are colored according to the measurement. The dots are colored according to the identity of primordia that were formed at the SAM periphery. Significant differences for each measurement among genotypes were determined via one-way ANOVA (two-sided), followed by Tukey post-hoc comparisons (*p* < 0.05). Data sets that share a common letter do not differ significantly. See Supplementary Data 7 for precise sample size and *p*-values of Mann-Whitney-Wilcoxon and ANOVA test. Source Data are provided as a Source Data file.

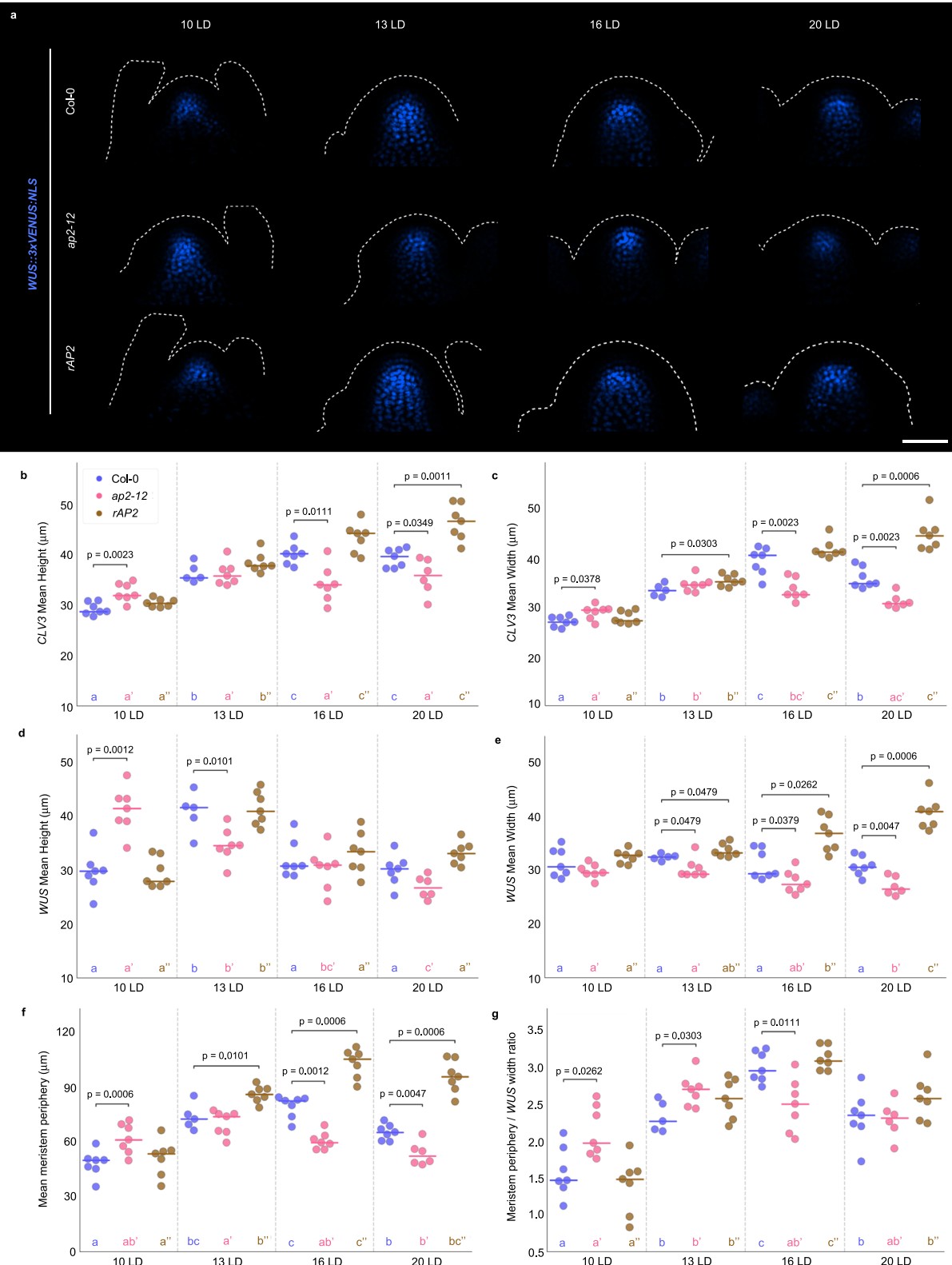

(Figs. 3c, e, f). Nevertheless, despite the increase in width of individual domains, at 20 LD the ratio between OC and PZ width is similar to that in Col-0 (Fig. 3g).

Overall, comparisons of the domains of the *ap2-12* mutant, *rAP2* and Col-0 SAMs through floral transition suggest that AP2 is required for the increase in height and width observed in the CZ of the inflorescence SAM compared to the vegetative SAM, and is required to increase the width of the PZ as well as to maintain the width of the OC.

Moreover, reduction of AP2 during floral transition ensures that the CZ, OC and PZ do not increase excessively in width.

## Mutual repression of *SOC1* and *AP2* contributes to the regulation of SAM morphology during floral transition

AP2 promotes SAM height and width, and delays floral transition. To understand in more detail how AP2 regulates these processes, global gene expression analysis was performed by RNA sequencing

**Fig. 3 | AP2 affects central zone and peripheral zone during floral transition.**
**a** Pattern expression of *WUS::3xVENUS-NLS* at Col-0, *ap2-12* and *rAP2* SAMs of
plants grown under continuous longs days (LDs). Each SAM is shown from the
side. The outline of each acquired meristem and its peripheral organs is indi-
cated with a dotted white line. Scale bar = 50 μm. **b**–**c** Quantification of the size of
*CLV3::mCHERRY-NLS* domain (**b**) height and (**c**) width in Col-0, *ap2-12* and *rAP2*
backgrounds. **d**–**e** Quantification of the size of *WUS::3xVENUS-NLS* domain (**d**)
height and (**e**) width in Col-0, *ap2-12* and *rAP2* backgrounds. **f** Mean meristem
periphery calculated subtracting *WUS* domain width (Fig. 3e) to the meristem
width (Supplementary Fig. 6c) and dividing by 2 that difference. **g** Ratio of

meristem periphery and *WUS* domain width in Col-0, *ap2-12* and *rAP2* back-
grounds. The horizontal bars represent the median value for each genotype.
Significant differences among genotypes within each time point were deter-
mined via two-sided Mann-Whitney-Wilcoxon test with the wild type and across
the same genotype at different time points via one-way ANOVA (two-sided),
followed by Tukey post-hoc comparisons ($p < 0.05$). For the ANOVA test, data
sets that share a common letter do not differ significantly. The color of the dots
and the letters correspond to the genotype. See Supplementary Data 7 for pre-
cise sample size and *p*-values of Mann-Whitney-Wilcoxon and ANOVA test.
Source Data are provided as a Source Data file.

(RNA-Seq) using apices of Col-0 and *ap2-12* mutants grown under
continuous LDs for 10, 12, 14 and 17 days. This approach yielded 103
genes that were differentially expressed (DEGs) in *ap2-12* compared to
Col-0 at one or more time point (Supplementary Data 1). To identify
genes involved in SAM morphology regulation, DEGs were identified at
14 LD by comparing the smaller *ap2-12* SAM to the elongated Col-0
SAM (Supplementary Data 2). These DEGs were then cross-referenced
with those identified by comparing the Col-0 SAM at maximum height
(14 LD) with the SAM before the height and width increased (10 LD)
(Supplementary Data 3). The list of DEGs common to both compar-
isons (Supplementary Data 4) was then compared with AP2 direct
targets identified by ChIP-Seq analysis[43] (Supplementary Data 5). This
approach filtered three direct targets of AP2 whose expression corre-
lated with SAM size during floral transition (Fig. 4a, b): *LIPOXYGENASE
2, COPPER AMINE OXIDASE ALPHA 2* and *SOC1*. Of these three genes,
only *SOC1* was more highly expressed in *ap2-12* at more than one time
point, and this gene was previously reported to regulate floral transi-
tion and SAM size regulation[13,20,47], so we focused on deciphering the
relationship between SOC1 and AP2 in controlling SAM shape and
primordium identity during floral transition.

The transcriptome analysis was first extended by comparing the
temporal and spatial patterns of SOC1 protein accumulation during
floral transition in *ap2-12* and Col-0 SAMs. The *SOC1::SOC1:GFP*
reporter[20] was analyzed in the *soc1-2* and *soc1-2 ap2-12* mutant back-
grounds (Figs. 4c, d; Supplementary Figs. 7a–b). At 10 LD and 12 LD,
SOC1:GFP accumulated more in the *ap2-12* SAMs (*ap2-12 soc1-2
SOC1::SOC1:GFP*) than in wild-type plants (*soc1-2 SOC1::SOC1:GFP*),
indicating that AP2 represses *SOC1* transcription in the wild-type SAM
during vegetative development. However, by 14 LD, SOC1:GFP abun-
dance had increased in wild-type SAMs and was present at a similar
level to that in *ap2-12* SAMs of *ap2-12* mutants (Figs. 4c, d). Together
with the direct binding of AP2 to the *SOC1* promoter (Supplementary
Figs. 7c, d)[43], these results suggest that AP2 directly represses *SOC1*
expression at the SAM before floral transition.

To test whether *SOC1* repression by AP2 contributes to SAM
morphology regulation, the SAMs of *soc1-2 ap2-12* double mutants
were compared to those of *soc1-2* and *ap2-12* single mutants during
floral transition. The maximum values of SAM height, width and area
for *soc1-2* SAMs were detected at 17 LD, 3 days later than in Col-0
(Figs. 5a–c, Supplementary Fig. 8). The maximum height of *soc1-2*
SAMs was similar to that of Col-0, but the maximum width was greater
(Figs. 2d, 5c). These results indicate that SOC1 hastens the increases in
SAM height and width that occur in Col-0, and reduces the final width
of the Col-0 SAM. The SAM shape of the *soc1-2 ap2-12* double mutant
was then analyzed (Fig. 5). At 10 and 12 LD, the width and height of *soc1-
2 ap2-12* and *ap2-12* SAMs were similar, suggesting that the increased
SOC1 expression detected at these time points (Fig. 4d) does not
immediately affect *ap2-12* SAM morphology (Figs. 5b, c; Supplemen-
tary Fig. 8d). However, at 14 LD, the *soc1-2 ap2-12* SAM was larger in
height and width than the *ap2-12* SAM, and similar to Col-0. These
results indicate that SOC1 activity reduces the height and width
of the *ap2-12* SAM at 14 LD. At 17 LD, the *soc1-2* mutant reached max-
imum height and width, and at this stage the SAM width of the *soc1-2
ap2-12* double mutant was still larger than that of *ap2-12* ($p < 0,0001$,

Mann-Whitney test) and comparable to Col-0. However, the height of
*ap2-12* and *ap2-12 soc1-2* SAMs at 17 LD were comparable. Overall,
analysis of the *soc1-2 ap2-12* double mutant suggests that SOC1 activity
contributes to the reduction of *ap2-12* SAM width at 14 and 17 LD, but
that it only transiently contributes to the reduction in SAM height at 14
LD ($p < 0,0001$, Mann-Whitney test). To test the significance of
increased SOC1 activity on SAM shape, *35S::SOC1:9xMYC* plants[48] were
analyzed. The height and width of *35S::SOC1:9xMYC* SAMs were
reduced throughout floral transition from 12 LD to 17 LD (Fig. 5d,
Supplementary Fig. 9), consistent with increased SOC1 expression in
*ap2-12* mutants altering SAM shape.

Comparison of *soc1-2 ap2-12* and *soc1-2* SAMs suggested that AP2
activity might contribute to the increased SAM height and width of the
*soc1-2* mutant. SOC1 has been described as a direct transcriptional *AP2*
repressor (Supplementary Fig. 7d)[20,21], suggesting that increased AP2
might contribute to the altered morphology of the *soc1* SAM during
floral transition. To test the effect of SOC1 on AP2 protein level,
*AP2::AP2:VENUS* expression was compared in *ap2-12* and *soc1-2 ap2-12*
SAMs during floral transition (Figs. 6a, b, Supplementary Fig. 10). As
previously described (Figs. 2a–c), in the *ap2-12 AP2::AP2:VENUS* back-
ground, the level of AP2:VENUS slowly decreased from 7 LD, but was
still present at 14 LD when Col-0 SAM reached maximum height, and
reached its lowest levels from 17 LD to 19 LD. In the *soc1-2 ap2-12
AP2::AP2:VENUS* mutant background, the temporal pattern of AP2:VE-
NUS was similar to that of *ap2-12 AP2::AP2:VENUS* background, but
AP2:VENUS was highly expressed in the SAM from 17 LD to 21 LD.
Therefore, AP2 protein accumulates for longer in *soc1-2* and at higher
levels compared to wild-type from 17 LD when the *soc1-2* SAM height
and width are increased compared to Col-0 (Figs. 5b, c). The sig-
nificance of increased AP2 expression at 17 LDs is also supported by the
reduced height and width of *soc1-2 ap2-12* SAMs at this time point
compared to *soc1-2* (Figs. 5b, c).

Transcriptional repression of *AP2* by SOC1 contributes to the
reduction in AP2 levels in the SAM during floral transition, but post-
transcriptional regulation by miR172 also represses AP2[40,49], and *rAP2-V*
increases SAM height and width towards the end of floral transition
(Figs. 1a, b, Supplementary Figs. 2e–f). To assess the effects on SAM
morphology of reducing both regulation by miR172 and SOC1, the
*rAP2-V soc1-2* line was constructed. The maximum width and height
of the *rAP2-V soc1-2* SAM was greater than of *rAP2-V* and *soc1-2*
(Figs. 5e, Supplementary Figs. 9c–d), demonstrating an additive effect
of transcriptional and post-transcriptional repression of AP2 on SAM
morphology.

## Mutual repression of *SOC1* and *AP2* affects flowering time and floral primordium identity

The mutual repression of *AP2* and *SOC1* regulates SAM shape during
floral transition, and AP2 promotes SAM size whereas SOC1 reduces it
(Figs. 1, 2d, 3, 5). However, these genes also have antagonistic effects
on flowering time with AP2 acting as floral repressor[43] and SOC1 as
floral promoter[13]. We therefore analyzed their mutual effects on flow-
ering time and primordium identity during floral transition. We first
compared days to bolting and number of rosette and cauline leaves
formed by *soc1-2*, *ap2-12* and *soc1-2 ap2-12* (Supplementary Figs. 8e–h),

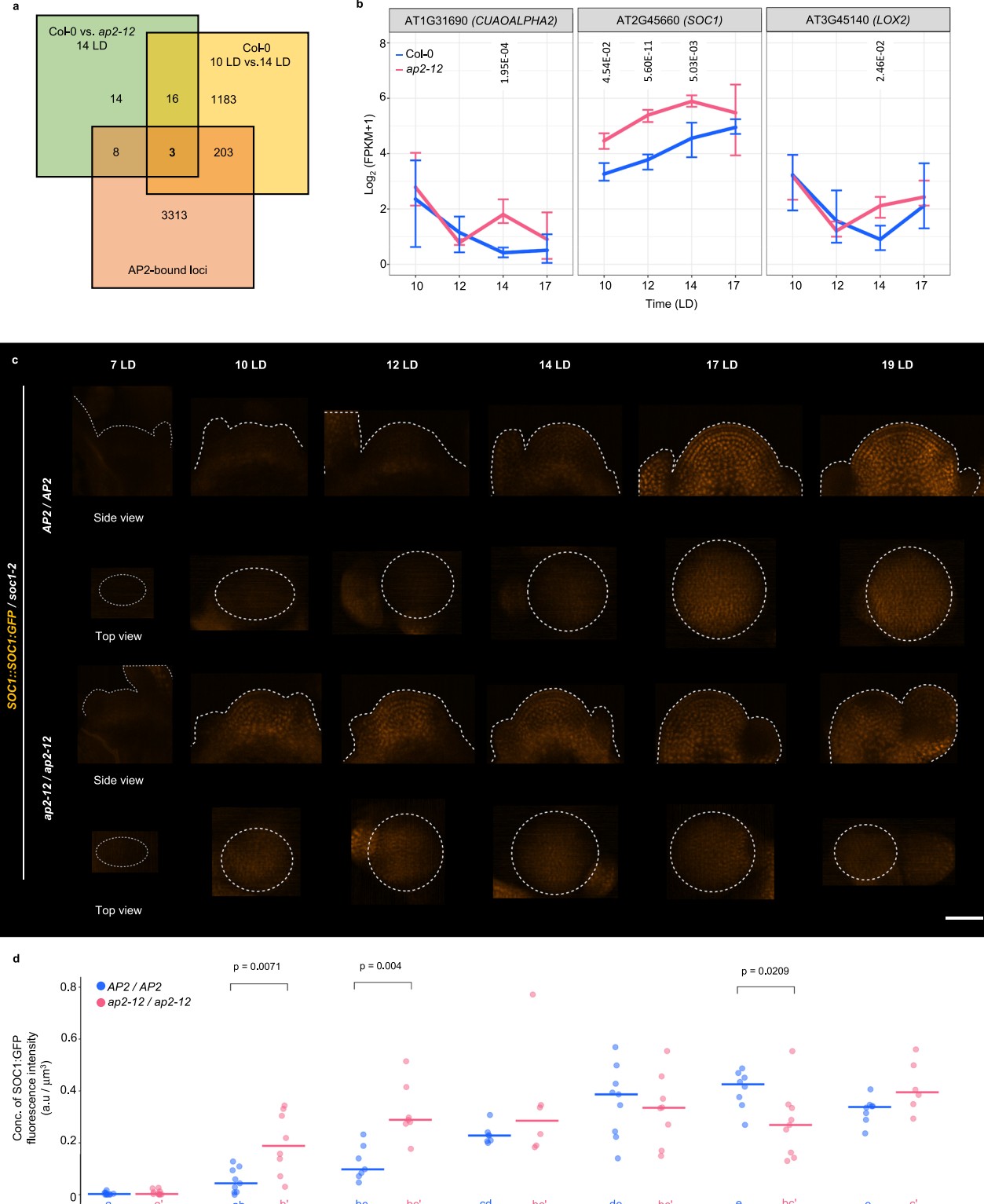

thereby extending previous data for total leaf number of *soc1-2 ap2-12*[43]. The *soc1-2 ap2-12* double mutants formed on average 6.2 more rosette leaves than *ap2-12* and Col-0, but 5.2 fewer than *soc1-2*, and bolted 2 days later than Col-0, at a similar time to *soc1-2*. Similarly, floral primordia were visible at *soc1-2 ap2-12* SAMs around 3 days later than *ap2-12* and two days earlier than *soc1-2* (Supplementary Fig. 8d). The *ap2-12* and *soc1-2* inflorescences formed fewer and more cauline

leaves than Col-0, respectively, whereas these organs were comparable in number in *soc1-2 ap2-12* and Col-0 (Supplementary Fig. 8f). Collectively, these data indicate that SOC1 is an important flowering promoter downstream of AP2, and that AP2 and SOC1 have antagonistic effects on cauline leaf number.

Similarly, *rAP2-V soc1-2* showed additive effects on SAM height and width compared to the single mutants, so the combined effects of

**Fig. 4 | AP2 is a negative regulator of *SOC1* expression at the SAM before and during floral transition. a–b** Global transcriptome profiling via RNA-Seq of Col-0 and *ap2-12* dissected meristems. **a** Venn diagram showing the overlap between the list of differentially expressed genes (DEGs) at 14 long days (LDs) between Col-0 and *ap2-12*, the list of DEGs in Col-0 between 10 LD and 14 LD, and the AP2-bound loci[43]. **b** Expression profiles under LDs for dissected plant apices of the genes that are present in the three lists compared in a. Error bars represent the range between the maximum and minimum values among the three replicates. Significant differences between genotypes at the same time point were determined via two-sided like-lihood ratio test (adjusted *p*-value < 0.05). **c** Pattern of protein accumulation of *SOC1::SOC1:GFP* at the SAM in *soc1-2* (AP2/AP2) and *soc1-2 ap2-12* (ap2-12/ap2-12) under LDs. In the side views, the shape of the acquired meristem and its peripheral organs is indicated with a dotted white line. In the top views, the orthogonal pro-jection on xz plane of the same meristem is shown (projection of 50 μm from the

top) and the meristematic region was highlighted using a dotted line. Scale bar = 50 μm. **d** Quantification of SOC1:GFP concentration of fluorescence intensity (total fluorescence divided by volume) at the shoot apex (from the tip to 50 μm deep in the basal direction) in *soc1-2* and *soc1-2 ap2-12* mutants during LDs. The horizontal bars represent the median value. Comparisons within each time point between genotypes were performed via two-sided Mann-Whitney-Wilcoxon-test ($p < 0.05$). Significant differences among time points within each genotype were determined via one-way ANOVA (two-sided), followed by Tukey post-hoc comparisons ($p < 0.05$). Data sets that share a common letter do not differ significantly. The color of the dots and the letters correspond to the genotype. The quantification of concentration of fluorescence intensity is consistent when performing the analysis with a depth of 20 μm, as shown in Supplementary Fig. 7b. See Supplementary Data 7 for precise sample size and *p*-values of Mann-Whitney-Wilcoxon and ANOVA test. Source Data are provided as a Source Data file.

*rAP2-V* and *soc1-2* on flowering time were also measured (Supple-mentary Figs. 9e–i). The *rAP2-V soc1-2* line showed a dramatic increase in rosette leaves, cauline leaves and days to bolting compared to each parental line (Supplementary Figs. 9e–g). Indeed, *rAP2-V soc1-2* formed on average 85.9 rosette and 51.4 cauline leaves compared to 26.2 and 6.0 for *soc1-2*, respectively. Moreover, the production of floral pri-mordia at the SAM was delayed compared to *rAP2-V* or *soc1-2* (Fig. 5e). These results demonstrate that *rAP2-V* and *soc1-2* showed additive effects on flowering time as well as SAM morphology.

## Discussion

We quantified the increases in SAM height and width that take place during floral transition in response to LDs, and showed that increases in CZ height and width, and PZ width, develop during floral transition and persist into the inflorescence SAM. We found that AP2 is present in the SAM during floral transition and is required for the large differ-ences in SAM height and width observed in Col-0, and that miR172 and SOC1 act additively to repress *AP2* at the end of floral transition to prevent excessive increases in SAM width and height. Moreover, we demonstrated that AP2 and SOC1 show reciprocal temporal patterns of expression in the SAM. This observation combined with the ability of both proteins to bind directly to each other's promoter[20,21,43] suggests that mutual repression of AP2 and SOC1 in the SAM plays a role in integrating SAM morphological changes with the acquisition of floral identity.

AP2 promotes SAM height and width during floral transition and represses flowering time. The effect of AP2 on SAM morphology is tightly temporally regulated, because it most strongly promotes changes in SAM height and width during the early stages of floral transition, although it is more strongly expressed earlier in the vegetative SAM. This observation suggests that the effect of AP2 on SAM morphology requires other factors that are active during floral transition. We find that AP2 is required for the increase in height and width of the CZ that occurs during floral transition and persists into the inflorescence SAM, and is required to maintain the width of the OC as well as to increase the width of the PZ. Moreover, reduction of AP2 during floral transition by miR172 is primarily necessary to ensure that the CZ, OC and PZ do not excessively increase in width soon after floral transition. *WUS*, which was used as the marker for the OC and encodes a homeobox transcription factor required for meristem maintenance[25], was previously found to be increased in expression by AP2 in floral meristems towards the end of floral development[50] and in the SAM at the end of inflorescence development when shoot growth has terminated[36]. *WUS* expression can be influenced by many mer-istem regulators[8,24,26–30], and some of the effects of AP2 on WUS expression might be caused indirectly by increasing the size of the OC, as we observed in rAP2 plants after floral transition. Similarly, rAP2 can promote inflorescence meristem size when expressed either in the *WUS* or *CLV3* domains[8], and AP2 was proposed to have a more complex role in maintaining SAM size by repressing CLV3 signaling or

increasing *WUS* expression[23]. Moreover, in the floral SAM, AP2 reg-ulates *WUS* indirectly through AUXIN RESPONSE FACTOR3[51]. The mechanism by which AP2 increases *WUS* expression and regulates SAM morphology is therefore likely to be complex and may involve several processes[52], but our analyzes show that repression of *SOC1* transcription is one way in which AP2 regulates SAM shape.

SOC1 promotes flowering, and in the *soc1-2* mutant SAM height increases more slowly during floral transition, perhaps due to the slower progression of the flowering program. However, in *soc1-2* mutants SAM width increases more rapidly towards the end of floral transition, and the inflorescence SAM becomes wider than the Col-0 inflorescence SAM. Analysis of the *ap2-12 soc1-2* double mutant, showed that AP2 is required for the *soc1-2* SAM to become wider than the Col-0 SAM, suggesting increased AP2 expression contributes to this increase in width. Accordingly, the *rAP2* transgene also increases SAM width. Nevertheless, the *soc1-2 ap2-12* SAM is still wider than that of *ap2-12*, indicating that SOC1 must also repress SAM width inde-pendently of AP2. Constitutive *SOC1* overexpression reduced SAM width and height, confirming the role of SOC1 as an inhibitor of SAM growth. SOC1 represses the expression of GA biosynthetic enzyme GA20ox2 at the SAM and increased GA levels may contribute to the wider SAM in *soc1-2*[5] and may explain the increment in width of *soc1-2 ap2-12* SAM compared to *ap2-12*.

The reciprocal repression of AP2 and SOC1 also contributes to the coordination of alterations in SAM shape with changes in primordium identity. During vegetative development, AP2 delays floral transition[43], in part by repressing *SOC1* transcription, but does not detectably influence vegetative SAM morphology (Fig. 6c). However, exposure of plants to LDs overcomes the repression of *SOC1* by AP2 (Figs. 6b, d), and a rise in SOC1 abundance represses *AP2* transcription (Fig. 6e). AP2 protein levels are also reduced through post-transcriptional regulation mediated by miR172[49,53], and our analysis of *rAP2-V* indicates that insensitivity to miR172 extends the duration of AP2 expression into the mature inflorescence SAM. SOC1 and miR172 additively repress AP2, as suggested by our analysis of *rAP2-V soc1-2* plants which showed strongly enhanced delayed flowering and larger SAMs. However, in the short time interval between the initiation of floral transition and the disappearance of AP2 from the SAM through the action of miR172 and SOC1, at about 12–14 LDs after germination, AP2 promotes increases in SAM height and width (Fig. 6d). Therefore, the doming of the SAM associated with flowering is limited to the early stages of floral tran-sition prior to the reduction in AP2 by the action of SOC1 and miR172, and while leaf primordia are still visible at the shoot apex. In *soc1* mutants, floral transition and the repression of *AP2* are both delayed, and the increase in SAM height is delayed while the increase in SAM width is enhanced, partially through increased AP2 expression. Thus, the mutual repression of SOC1 and AP2 determines the timing of floral transition, ensures that the SAM rapidly increases in height and width during the early stages of floral transition and that it does not exces-sively increase in width.

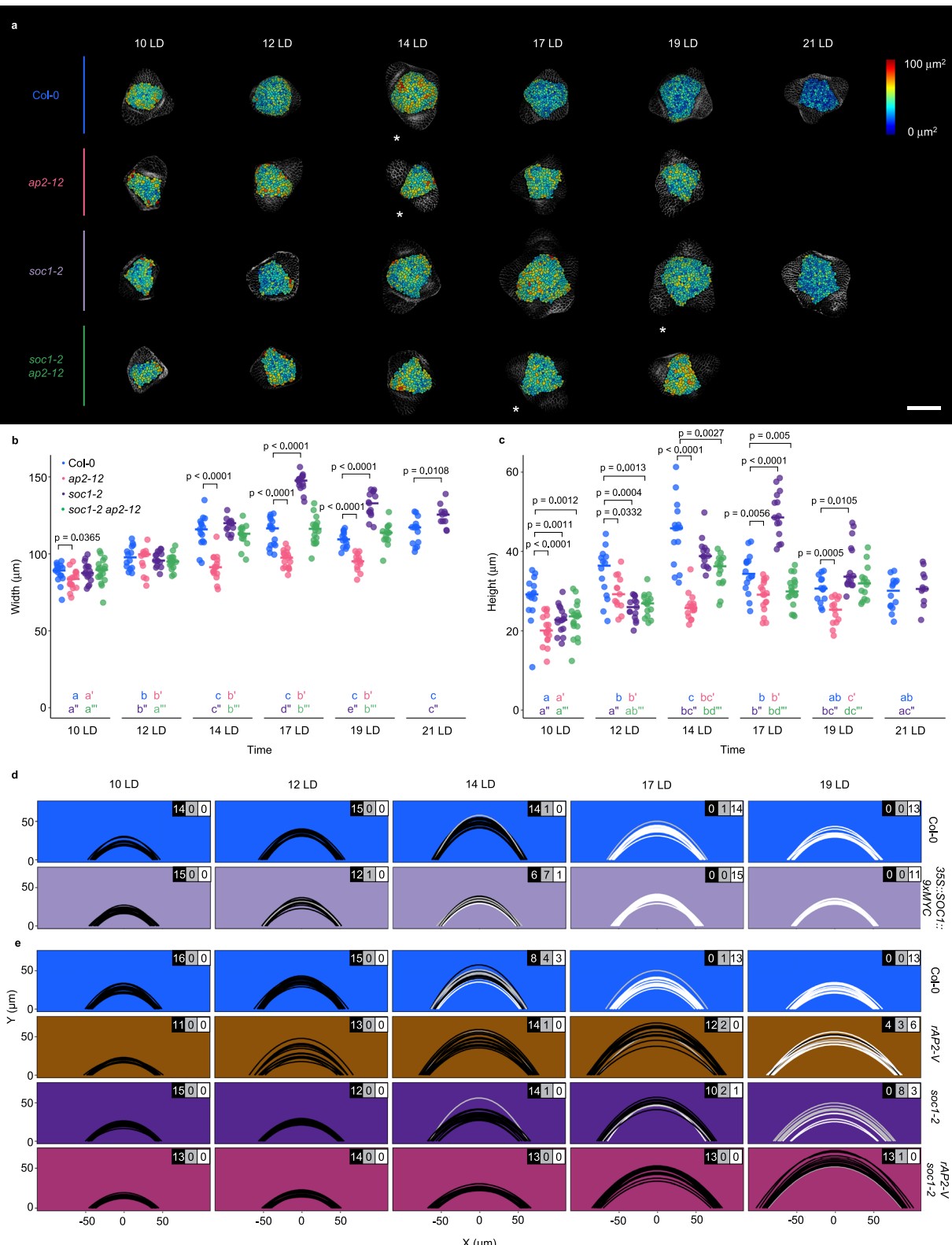

A feature of our model is the importance of mutual repression of *AP2* and *SOC1*. This type of direct reciprocal repressive motif has been characterized in developmental processes in animals[54–56], and is proposed to sharpen and steepen spatial boundaries of gene expression[57]. AP2 and SOC1 are expressed in a similar spatial pattern throughout the SAM, but show different temporal patterns, with AP2 being expressed before SOC1. Thus, the mutual repression of SOC1 and AP2 determines the time interval during which both factors are expressed, which limits the stage during which AP2 promotes changes in SAM morphology.

## Methods

### Plant material and growth conditions

All plants in this study were *Arabidopsis thaliana* Columbia-0 (Col-0) background. Mutant alleles were previously described: *ap2-12*[43] and

**Fig. 5 | Mutual repression of SOC1 and AP2 affects flowering time and floral primordium identity. a** Top view of the heatmap quantification of cell area in the meristem region via segmentation in SAMs of Col-0, *ap2-12*, *soc1-2* and *soc1-2 ap2-12*. White asterisks indicate the first time point at which floral primordia were detected in the analysis of the corresponding genotype. Scale bar = 50 µm. *n* = 4 SAMs. **b**–**c** Measurement of (**b**) width and (**c**) height of the SAM in continuous long day (LD)-grown plants. Significant differences between wild-type and mutants within each time point were determined via two-sided Mann-Whitney-Wilcoxon-test (*p* < 0.05). Significant differences among time points within each genotype were determined via one-way ANOVA (two-sided), followed by Tukey post-hoc

comparisons (*p* < 0.05). Data sets that share a common letter do not differ significantly. The color of the dots and the letters correspond to the genotype. **d**–**e** SAM morphology of (**d**) Col-0, *rAP2-V*, *soc1-2*, *rAP2-V / soc1-2* and (**e**) *35::SOC1:9xMyc* under continuous LDs. The parabolas are colored according to the identity of primordia that were formed at the SAM periphery. The number of meristems producing each kind of primordia are listed on the top-right corner in each genotype and time point. See Supplementary Data 7 for precise sample size and *p*-values of Mann-Whitney-Wilcoxon and ANOVA test. Source Data are provided as a Source Data file.

*soc1-2*[14]. The following transgenic lines were used: *AP2::AP2-VENUS #13*[40], *AP2::rAP2-VENUS #A6*[8] and *SOC1::SOC1-GFP*[20]. The *ap2-12 soc1-2* double mutant (previously published in ref. [43] and reconstructed here) and the *AP2::AP2-VENUS #13 soc1-2 ap2-12*, the *SOC1::SOC1-GFP soc1-2 ap2-12* and the *AP2::rAP2-VENUS #A6 soc1-2* genotypes were generated in this study via crossing. Plants carry both *CLV3::mCHERRY-NLS* and *WUS::3xVENUS-NLS*[4] were used to mark the CZ and OC and were crossed in *ap2-12* and *rAP2* line B2[8]. Genotyping was performed using the primers listed in Supplementary Data 6 or by performing the phosphinotricin (PPT) resistance assay (see below). Plants were grown on soil under controlled conditions of SDs (8 h light/16 h dark) and LDs (16 h light/8 h dark).

### PPT resistance assay

The identification of PPT-resistant plants was performed on agar plates using Murashige & Skoog (MS) medium containing PPT similar to[40], but using 1× MS medium containing 15 mg mL$^{-1}$ PPT (without sucrose), and placing the plates with leaves in continuous light at 21 °C for at least 5 days.

### Confocal imaging

Shoot apices at different developmental stages were dissected under a stereomicroscope and fixed with 4% (v/v) paraformaldehyde (PFA; Electron Microscopy Sciences). The fixed samples were washed twice for 1 min in phosphate-buffered saline (PBS) and cleared with ClearSee[58] for 3–4 days at room temperature. Before imaging lines containing a fluorescent reporter, samples were kept in PFA for 2 h at room temperature after fixation and were then transferred to PBS for 2 days and then to ClearSee for 3–4 days. The cell wall was stained with Renaissance 2200 [0.1% (v/v) in ClearSee][59] for at least 1 day.

Confocal microscopy was performed with a TSC SP8 confocal microscope (Leica) for cell segmentation and SAM morphology quantification, where Renaissance was excited at 405 nm and image collection was performed at 435–470 nm (Figs. 1, 2d, 5, Supplementary Fig. 1, Supplementary Fig. 2, Supplementary Fig. 4, Supplementary Fig. 5, Supplementary Figs. 8a–d and Supplementary Figs. 9a–d, Supplementary Figs. 11a–b). The protein patterns of AP2:VENUS, rAP2:VENUS and SOC1:GFP at the SAM were acquired with a Stellaris 5 confocal microscope (Leica) for fluorescence quantification (Figs. 2a–c, 4c, d, 6a, b, Supplementary Fig. 3, Supplementary Figs. 7a–b, Supplementary Fig. 10, Supplementary Fig. 12). VENUS and GFP were excited at 515 nm and 488 nm, and the signal was detected at 520–600 nm and 500–557 nm, respectively. The spatial patterns of expression of *WUS::3xVENUS-NLS* and *CLV3::mCHERRY-NLS* were acquired as well with a Stellaris 5 confocal miscroscope (Fig. 3, Supplementary Fig. 6, Supplementary Fig. 11). VENUS and mCHERRY were excited at 515 nm and 587 nm and the signal was detected at 520–540 nm and 600–620, respectively. For all time courses where protein accumulation patterns were determined, Renaissance signal was detected using similar parameters as mentioned earlier for the segmentation analyzes.

### Cell segmentation and SAM morphology quantification

The z-stacks of SAMs were acquired with a step size of 0.4 µm and were converted to TIF files with Fiji. MorphoGraphX (MGX) software

(https://morphographx.org/)[60,61] was used to extract the surface of the meristem and to project the Renaissance signal of the cell wall from the outer cell layer (i.e., L1), which was used to segment the images. Cells were segmented using the "auto-segmentation" function and corrected manually. The geometry of the surface was displayed as Gaussian curvatures with a neighboring radius of 10 µm. The boundary between the meristem and the developing primordia was defined by a negative Gaussian curvature, then the area of each of the cells in the SAM was extracted. The meristem area was calculated as the sum of the areas of the cells that comprised the meristem.

To quantify the morphology of the meristem, its height and width were estimated (Supplementary Figs. 11a–b). For this, the orthogonal views from the z-stacks were generated and were used to estimate the meristem height and the width according to the following criteria: (1) the height aligned with the apical–basal axis, (2) the width was perpendicular to the height and (3) the width was measured from the most apically visible primordium. The measurements performed on each orthogonal view were considered as technical replicates; thus, the plotted values corresponded to the means of the two estimations of each of the measured parameters. The parabolas to represent meristem morphology were fitted in an XY-coordinate system (Supplementary Fig. 11b) using the formula in Supplementary Fig. 11b. For representation purposes, the parabolas were colored according to the identity of primordia that were formed at the SAM periphery (Figs. 1d, 5d, e, and Supplementary Fig. 2g, Supplementary Fig. 5, Supplementary Fig. 8) as shown in Supplementary Fig. 4. For the time curse of Col/0 vs. *sup-41*, acquisitions from the side were performed. Therefore, the most central slice was selected, and then height and width measurements were performed according to the aforementioned criteria.

### Fluorescence quantification of *SOC1-GFP* and *AP2-VENUS*

Confocal fluorescence z-stacks with *SOC1-GFP* and *AP2-VENUS* were processed and analyzed using a Matlab custom-made code (https://gitlab.com/slcu/teamHJ/pau/RegionsAnalysis), which was adapted and extended with the pipelines presented below. The main goal of this analysis was to extract reproducible measures of fluorescence intensity within the SAM. A normalized fluorescence intensity measure was computed as a proxy for the concentration of protein at the meristem upper region. To do that, a semi-automatic pipeline was developed, which is described as follows.

Due to the difference in the resolution between the xy plane and the z-direction (depth), the z-stack was resized by increasing the number of slices in the z direction through bicubic interpolation to obtain a homogeneous volumetric resolution.

To exclude fluorescence signals outside the region of interest and quantify only the fluorescence intensity within the meristematic region, a pre-processing step was performed: a 3D paraboloid mask was constructed using the curvature of the meristem (Supplementary Fig. 12). First, a stack-slice interval that contained the apex of the meristem was selected and the cell wall signal present within this interval was projected in each orthogonal plane (xy and yz) (Supplementary Fig. 12a). Then, two curved lines following the parabolic outline of the SAM were drawn in the xy and yz planes, in the sum of slice projection of each plane (Supplementary Fig. 12b). Later, a

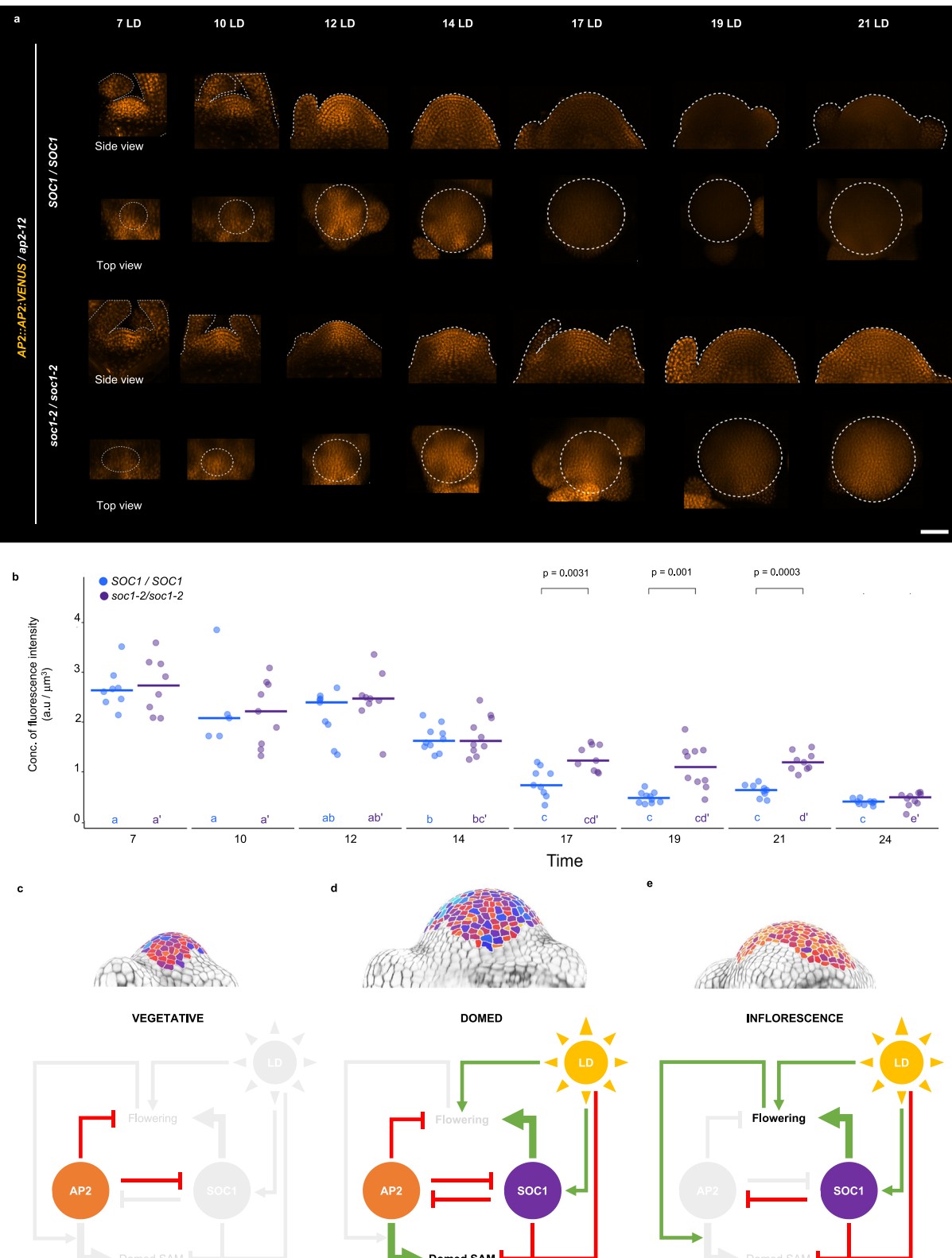

parabolic fitting of the two drawn lines was performed (Supplementary Fig. 12c), which considered a potential tilting of the SAM with respect to the vertical axis. Specifically, the code recursively fits parabolas in different orientations of the drawn outline, and choses the one that minimizes the $R^2$ value. From the two orthogonal parabolas fitted for each z-stack, the apex was computed to derive the equation for the 3D paraboloid. The $z_o$ coordinate of the paraboloid was determined from averaging the apices of the orthogonal parabolas (Supplementary Fig. 12d). The parameter $a$ in the parabola equation, i.e the curvature, was used to substitute the denominator terms in the paraboloid equation ($c_1^2$ and $c_2^2$; Supplementary Fig. 12f) so that the paraboloid equation matched the linear and quadratic terms of each of the equations at $y = y_o$ and $x = x_o$, respectively (Supplementary Fig. 11f). Because the z-stack did not always include the beginning and end of

**Fig. 6 | SOC1 is a negative regulator of *AP2* expression at the SAM during floral transition. a** Pattern of protein accumulation of *AP2::AP2:VENUS* at the SAM in *ap2-12 (SOC1/SOC1)* and *soc1-2 ap2-12 (soc1-2/soc1-2)* plants under continuous longs days (LDs). In the side views, the outline of each acquired meristem and its peripheral organs is indicated with a dotted white line. In the top views, the orthogonal projection on xz plane of the same meristem is shown (projection of 50 μm from the top) and the meristematic region was highlighted using a dotted line. Scale bar = 50 μm. **b** Quantification of AP2:VENUS concentration of fluorescence intensity (total fluorescence divided by volume) at the shoot apex (from the tip to 50 μm deep in the basal direction) in *ap2-12* and *soc1-2 ap2-12* mutant backgrounds during continuous LDs. The dots are colored according to the mutant background of the analyzed plant. The horizontal bars represent the median value for each genotype.

Comparisons within each time point between genotypes were performed via a two-sided Mann-Whitney-Wilcoxon-test ($p < 0.05$). Significant differences among time points within each genotype were determined via one-way ANOVA (two-sided), followed by Tukey post-hoc comparisons ($p < 0.05$). Data sets that share a common letter do not differ significantly. The color of the dots and the letters correspond to the genotype. The quantification of the concentration of fluorescence intensity is consistent when performing the analysis with a depth of 20 μm, as shown in Supplementary Fig. 10c. See Supplementary Data 7 for precise sample size and *p*-values of Mann-Whitney-Wilcoxon and ANOVA test. Source Data are provided as a Source Data file. **c**–**e** Schematic representation of AP2 and SOC1 regulation of SAM morphology and flowering at the (**c**) vegetative, (**d**) floral transition, and (**e**) inflorescence stages.

the meristem in the yz plane (lateral view), only the xy curvature was used ($c_2^2 = c_1^2$). Finally, to extract the fluorescence signal within the SAM using the 3D paraboloid, a 2D parabolic mask was created for each of the z-stack slices, and, in each slice of the z-stack, all intensity values of the pixels outside the paraboloid were set to 0 (Supplementary Fig. 12d, g).

To exclude any fluorescence signal at the boundaries of the SAM and primordia, the paraboloid curvature was increased with respect to the original (Supplementary Figs. 12d, g) such as $a' = a/\alpha$, being $a'$ the curvature of the new paraboloid and α the image resolution (α < 1 μm) (Supplementary Fig. 12d).

A concentration of fluorescence intensity measure was computed as the fraction between the total intensity (sum of the voxels' intensity) and the total volume (sum of the voxels' volume) in the upper region of the paraboloid with increased curvature. This upper paraboloid region was delimited between the 3D paraboloid itself and a transversal plane set at a distance of 20 μm or 50 μm from the paraboloid apex (Supplementary Fig. 12e). An example of quantified region within the paraboloid in a single confocal slice is shown in Supplementary Fig. 12g. For *SOC1::SOC1-GFP/soc1-2 ap2-12* fluorescence quantification a Gaussian filter (sigma = 2.5) was applied in the region within the paraboloid. For representation purposes, the values for the concentration of fluorescence intensity were divided by 1000.

**Fluorescence quantification of *WUS::3xVENUS-NLS* and *CLV3::mCHERRY-NLS***

Confocal fluorescence z-stacks showing the *WUS* and *CLV3* transcriptional reporters were also processed and analyzed using Matlab custommade code (https://gitlab.com/slcu/teamHJ/pau/RegionsAnalysis)[3,62]. The main goal of this analysis was to extract reproducible measures of the regions at the SAM with higher gene expression of *WUS* and *CLV3*. This analysis was performed using a semi-automatic pipeline represented in the Supplementary Figs. 11c–h.

First, the original stack (Supplementary Fig. 11c) is reduced into a sub-stack that only contains the central region where the SAM is located to exclude the fluorescence signal from the primordia. To obtain this working stack, the pipeline shows a top view of a maximal intensity projection of the desired fluorescent marker (Supplementary Fig. 11d). In this image, the user can select the rectangular region that contains the SAM. This will crop the original stack with the limits defined by the rectangle, as shown in the Supplementary Fig. 11e.

The program then performs the projection of the fluorescence signal in the two orthogonal directions by summing all the fluorescence signal across these directions, providing a 2D image for each orthogonal side view (Supplementary Fig. 11f). On those resulting projected images, we can define isoclines that delimitate regions with an expression higher than a certain signal level (Supplementary Fig. 11g). Finally, the program fits an ellipse to the desired isocline and assigns the two semiaxis to height and width comparing it with the vertical axis of the image (Supplementary Fig. 11h). For this analysis we have chosen the region that presents at least more than 50% the maximal expression observed, to ensure we are characterizing the regions in which *WUS* and *CLV3* are

highly expressed. Due to the dilution effect, this inherited signal would be much lower than the 50% after just a couple of rounds of division. For the quantification of the SAM periphery size, we subtracted the width of the *WUS* domain from the measured width of the meristem and divided this quantity by two. We chose the *WUS* domain as a reference to delimit the peripheral region because it is more aligned with the region where the primordia are created.

**Gene expression and whole-transcriptomic RNA-sequencing analysis**

Shoot apices of Col-0 and *ap2-12* mutants were dissected under a stereo microscope at 10, 12, 14 and 17 days in LD conditions in three independents biological replicates. Total RNA was extracted using the RNeasy Plant Mini Kit (Qiagen, USA) and subjected to DNase treatment using the TURBO DNase (Invitrogen). Poly(A) RNA enrichment, library preparation, and sequencing were carried out at the MPIPZ Genome Center, Cologne, Germany using the following conditions: The RNAs were processed by poly-A enrichment followed by application of basic components of "NEBNext Ultra II Directional RNA Library Prep Kit for Illumina" with a homebrew barcoding regime. Sequencing was performed on a HiSeq3000 sequencer by sequencing-by-synthesis with $1 \times 150$ bp single-read length. Sequence reads were preprocessed to remove any residual adapters with CutAdapt, and the low-quality bases (Q < 15) were trimmed from the ends with Trimmomatic[63,64]. Only reads with a minimum length of 50 nucleotides were kept. Salmon was used to quantify the abundance of transcripts from the Arabidopsis reference genome Reference Transcript Dataset for Arabidopsis (including guanine/cytosine bias, unstranded samples)[65,66]. Fragments Per Kilobase of transcript per Million (FPKM) values and corrected *p*-values were obtained using DESeq2 by comparing Col-0 to *ap2* in each time point using standard settings. Differentially expressed genes (DEGs) were defined in each comparison via DESeq2[67] (i.e., adjusted *p*-value < 0.05 and absolute $\log_2$ Fold Change > 1).

**Statistics and reproducibility**

No statistical method was used to predetermine sample size. Plants from one experiment (or one biological replicate in the RNA-Seq) grew together in the same chamber. The position of each plant in the space allocated in the growing chamber was randomized to avoid biases due to non-homogeneous growing conditions. To avoid possible effects of the clearing treatment in the quantified fluorescence, samples from each genotype were included in each imaging session when the acquisition of one time point lasted more than one day. Due to either defects in the meristem sample (i.e breakage of the tissue or leaf primordia blocking the fluorescence signal) and/or strong developmental differences with respect to the rest of the samples within the same time point, some meristems were not considered for fluorescence quantification. Severely damaged meristems were not considered for morphology analyzes. Among all the imaged meristems, 4 SAMs per time point and genotype were selected for MorphoGraphX analysis. The meristems were selected by absence of damage, absence of developing organs covering the meristem and overall quality of the image.

When more than 4 meristems were suitable for MorphoGraphX analysis, the four first imaged meristems were arbitrarily selected. The investigators were not blinded to allocation during experiments and outcome assessment, because the used material in this study was rigorously labeled, thus making blinding not possible.

## Reporting summary

Further information on research design is available in the Nature Portfolio Reporting Summary linked to this article.

## Data availability

The raw data from RNA-Seq series generated in this study have been deposited in NCBI Sequence Read Archive database under the accession code PRJNA954448. Other raw data and the original confocal microscope images are available on Edmond database under the accession code 3.G0AEP5_2024 (https://doi.org/10.17617/3.G0AEP5). Source data are provided with this paper.

## Code availability

Source code for the image analysis has been deposited in GitHub [https://gitlab.com/slcu/teamHJ/pau/RegionsAnalysis]

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

## Acknowledgements

We thank C. Ferrándiz and the members of the laboratory of G.C. for engaging scientific discussions. We thank J. Lohmann for sharing the *WUS* and *CLV3* transcriptional reporters. We thank J. Chandler for critical reading of the article. We are grateful to B. Huettel and the staff from the Max Planck Genome Center for their support during the RNA-sequencing experiment. We thank the staff from the cultivation facilities of the MPIPZ. M.C. received a post-doctoral fellowship from the Alexander von Humboldt foundation and K.W. received a studentship from the China Scholarship Council. GC receives funding from the Deutsche Forschungsgemeinschaft (DFG) through grant CO 318/14-1. GC and PFJ also receive funding from the DFG through the Cluster of Excellence CEPLAS (EXC 2048/1 Project ID: 390686111) and from a Core Grant from the Max Planck Society.

## Author contributions

E.B.Gd.O., M.C., and G.C. conceived and designed the study. A.V., E.B.Gd.O., and E.S. produced the RNA-Seq data. E.B.Gd.O., G.R.-M., M.C, and Y.L.S. performed confocal imaging and the subsequent analyses. G.R.-M. and P.C.-F. designed the image quantification pipelines and performed the analyses of the fluorescence reporters. K.W. introduced the AP2 translational reporter in the *ap2-12 soc1-2* mutant background. Y.L.S. and S.S. provided technical support. A.V., P.F-J., and G.C. provided scientific guidance throughout the project. E.B.Gd.O., M.C., and G.C. wrote the manuscript. All the authors discussed the results and commented on the manuscript.

## Funding

## Competing interests

The authors declare no competing interests.
