## [Peer Review File · Nature Communications]

REVIEWER COMMENTS

Reviewer #1 (Remarks to the Author):

The manuscript by Garcia de Olalla and colleagues presents an elegant genetic mechanism controlling the morphology of the *Arabidopsis thaliana* shoot apical meristem during the transition from the vegetative to the reproductive phase. During this transition, the meristem is substantially enlarged forming a much more pronounced dome, before acquiring the much more stable shape and size of the inflorescence meristem. This is all the more remarkable, since meristem size and shape are tightly controlled and robust against even substantial perturbations. The authors make the case that AP2 is engaged in a cross negative regulatory interaction with SOC1, both transcription factors transiently expressed throughout the meristem and both with roles in regulating flowering time. They show convincingly that the transient co-expression of AP2 and SOC1 coincides with doming and that removing one of the factors largely suppresses this behavior. The manuscript presents highly convincing and beautiful data, but falls short of explaining how the two transcription factors cause such a dramatic morphological shift. What are the responsible cells? Does the SAM accumulate more stem cells, or is it rather additional cell proliferation at the periphery? Which cells respond to the AP2/SOC1 instruction? Does regionally restricted expression of the pair phenocopy the doming? Can you extend the time of doming by ectopic expression? What are the signaling mechanisms downstream of AP2 and SOC1, for example is cytokinin involved? Since many of the mechanisms controlling cell behavior in the meristem are known, I would very strongly encourage the authors to include some data on these aspects.

Minor point: Please include top views of the AP2 and SOC1 reporters to allow the reader to assess their 3D expression patterns.

Reviewer #2 (Remarks to the Author):

This manuscript dissects the role of AP2 in shoot doming during the developmental transitions that lead from the vegetative to the reproductive phase in *Arabidopsis*.

The first part of the paper focuses on the role of AP2 with section headings stating (1) AP2 positively regulates meristem area, (2) AP2 increases SAM height and width during the floral transition and (3) AP2 protein is present in the SAM during the floral transition.

Major concern for this section.

A. There is strong overlap between the data here and prior papers from the Coupland lab, primarily Sang et al 2022 (<https://doi.org/10.1111/nph.18111>) and for one aspect (AP2-VENUS accumulation during the floral transition) Ó'Maoiléidigh et al 2021 (<https://journals.plos.org/plosbiology/article?id=10.1371/journal.pbio.3001043>).

In particular Figures 1-3 in Sang et al correspond to data in Figures 1 and 2 in the current paper, except that the effect of AP1:VENUS and rAP2:VENUS accumulation was assayed during the floral transition in long days in Fig. 1 in Sang et al (day 12, day 15 (first flower)). In addition, Fig. 5 in Ó'Maoiléidigh showed that AP2-VENUS was still present in the SAM at day 14 in long days, that is to say during/after the floral transition. The authors use florally non-inductive short days plus photoperiod treatment here in a time course and come to the same conclusion (Fig. 2, heading 3 above). In Fig. 3, Sang et al. measured the meristem size, cell number and cell area in wild type, ap2 rAP2:VENUS plants in 1 cm bolted plants grown in long days, concluding that the meristem size and cell number are reduced in ap2 and increased in rAP2-VENUS relative to the wild type, while the cell area in the L1 area is increased in both ap2 and rAP2-VENUS. The same conclusions are drawn in Fig. 1 of the current manuscript for short day grown plants (heading 1 and 2 above). In Fig. 2 Sang et al. looked at changes in developmental timing in Col, ap2, AP2:VENUS, AP2:VENUS ap2, rAP2:VENUS, and rAP2:VENUS ap2 (rosette leaf number, rosette and cauline leaf number (= total leaf number), which is also reported in Fig. 1d in the current manuscript). Thus, major conclusions ascribed to the data in the current paper had already been reached in prior publications by the same lab.

B. The authors report that changes in meristem shape, in particular doming (vertical increases) are coupled to the floral transition by AP2 and SOC1 (see below). However, Fig. 1 d and Fig. S5 show clearly that the highest doming is first achieved around day 7 in all three genotypes tested. At this stage, the wild type has made many cauline leaves (end of 'transition stage'), ap2 is making flowers (is now in what the authors call the 'inflorescence stage'), while rAP2-VENUS is just starting to exit vegetative development. At d7 all genotypes have also reached maximum width except the rAP2 gain-of-function mutant, of all genotypes this one tested uniquely decouples doming (height increase) from width increase (Figure S4 and S6, current manuscript). The combined data argue strongly that doming and meristem size are not linked to developmental fate transition, they are decoupled – this contradicts the main conclusion and title of the paper.

Part two headings state (4) AP2 delays the floral transition by repressing SOC1 before floral transition and (5) The interaction between SOC1 and AP2 coupled changes SAM morphology to the floral transition.

Major concerns:

C. The authors identify SOC1 using genomic approaches as a candidate AP2 target based on elevated expression in *ap2* mutants and during the floral transition and the locus being bound by AP2. All these findings were known about SOC1 from prior genomic datasets (Matthieu 2009, Yant 2009, Schmidt 2003). Yant had also shown genetically that SOC1 acts downstream of AP2, but *soc1* is not fully epistatic to *ap2*. Moreover, the Coupland lab had linked SOC1 to promotion of doming via direct positive regulation of gibberellin biosynthesis (Kinoshita et al, 2020 <https://elifesciences.org/articles/60661>).

Using the exact same photoperiod shift setup as in the current paper, *soc1* mutants and gibberellin biosynthesis mutants show reduced meristem area already at + 5 long day (Kinoshita et al), prior to the stage when AP2 is reduced (+7 long day, current manuscript). Likewise, SOC1-GFP levels increase in wild type around day 12 in long days (Fig. 3 current manuscript), with AP2-VENUS still unchanged (Fig. 5b current manuscript), these data do not support heading 4.

Finally in wild-type morphology plants (AP2-VENUS rescues *ap2*, Sang et al), AP2 expression is fully downregulated only at day 17; Fig. 5b), while first flowers form in the wild type already at day 14 (Fig. 4b), inconsistent with the model proposed (Fig. 6).

D. As mentioned above in B for photoperiod shifts the change in meristem size and doming in are decoupled from the developmental transitions that lead to flower formation.

These data call in question whether the interaction between SOC1 and AP2 couples changes SAM morphology to the floral transition (heading 5).

Additional comments

1. For Fig 2, the authors should mention the number of samples they examined on each day. Or they could perform image quantification as they did for SOC1-GFP.

2. For Fig 3, SOC1-GFP in wild type background shows weaker expression compared to that in *ap2* background on day 10 and day 12, which does not match the phenotypic data where *ap2* has smaller meristem than WT from day 14 onwards (Supplementary Fig 5b). It seems SOC1 expression is released in *ap2* mutant only before day 14, while the phenotypic change in *ap2* mutant happens only after day 14.

3. To thoroughly test the mutual regulation between SOC1 and AP2, the SOC1 binding sites on AP2-Venus and mutating the AP2 binding sites on SOC1-GFP should be mutated. This excludes indirect effects the two known regulators of the floral transition and would be novel information.

4. Since there are differences in the duration of developmental stages in different lines, comparing the same fluorescent reporters in different backgrounds on the same day might just be comparing fluorescence reporters at different stages of plant development (e.g. Fig 2, 3d, 5a). This could be avoided using transient gain or loss of function.

5. For the image quantification method, the authors mentioned “the paraboloid curvature was increased with respect to the original such as $a' = a/\sqrt{2}$ ”. The author should show the exact value of $\sqrt{2}$ and how they determine the value. And it is better to show an example of quantified region in a single confocal slice in Supplementary Fig 11 (e.g. masking the final region of quantification in Supplementary Fig 11d).

Reviewer #3 (Remarks to the Author):

During the floral transition, SAM enlarges and gives rise to floral primordia instead of leaf primordia; however, how the meristem size and shape changes during the floral transition are regulated remains largely unknown. The manuscript is entitled, “A regulatory gene network that couples floral transition to shoot apical meristem morphology in Arabidopsis”. The authors found that SOC1 is activated during floral transition, and then represses the expression of AP2. AP2 is an essential regulator for meristem enlarge during the transition stage. The high level and the duration of AP2 expression enable a larger size of SAM. The authors establish a SOC1-AP2 feedback regulatory network to ensure the proper size of SAM during the floral transition. I have some questions and suggestions as mentioned below.

1. The size of the meristem greatly changes during the floral transition but the change mostly disappears in the ap2 mutant, indicating AP2 is an essential regulator for meristem enlarge during transition. As we know, the plants flowering early are more likely with a small meristem, and vice versa. The different meristem sizes with altered flowering times are all due to the changes in the expression level of AP2? What is the AP2 expression in other flowering-time mutants?

2. The flowering time of WT, ap2-12, soc1-2, and ap2-12 soc1-2 are all different. The meristem size changes a lot during the floral transition. So, I suggest the authors give a guideline on how to compare the meristem size of plants with different flowering time. The meristem size of ap2-12 soc1-2 is rescued. Is it because of the partially rescued flowering time?

3. The authors mention two growth conditions, one is “grown for 2 weeks under non-inductive short days (2 wSD) then exposure to LDs”, and the other is “grown under continuous LD”. I hope the authors will explain the reasons for setting these two conditions in detail.

4. The authors predict AP2 regulates meristem size during the floral transition most through increasing WUS transcription. Do authors examine the WUS expression in WT and AP2 during the transition stage?

5. What is the meristem phenotype of 35S:rAP2? It can rescue the meristem size of 35S:SOC1?

Minor points

1. Please explain why setting p value to 0.075 rather than 0.05 in Figure 5.

2. In Figures 1a,b, c, Why the ap2-12 data in +11LD is missing?

Response to reviewers

We thank the three reviewers for their interest in our work and their suggestions for how to improve the manuscript. The reviewers' comments appear verbatim below in italics, and our responses appear in bold.

Reviewer #1 (Remarks to the Author):

The manuscript by Garcia de Olalla and colleagues presents an elegant genetic mechanism controlling the morphology of the Arabidopsis thaliana shoot apical meristem during the transition from the vegetative to the reproductive phase. During this transition, the meristem is substantially enlarged forming a much more pronounced dome, before acquiring the much more stable shape and size of the inflorescence meristem. This is all the more remarkable, since meristem size and shape are tightly controlled and robust against even substantial perturbations. The authors make the case that AP2 is engaged in a cross negative regulatory interaction with SOC1, both transcription factors transiently expressed throughout the meristem and both with roles in regulating flowering time. They show convincingly that the transient co-expression of AP2 and SOC1 coincides with doming and that removing one of the factors largely suppresses this behavior. The manuscript presents highly convincing and beautiful data, but falls short of explaining how the two transcription factors cause such a dramatic morphological shift.

We appreciate the reviewer's comments on the quality of our data and the convincing demonstration of the significance of the co-expression of AP2 and SOC1. We understand the point that the mechanisms by which AP2 and SOC1 contribute to meristem shape changes were not addressed sufficiently and have provided additional data including the assessment of the role of AP2 in regulating the size of different meristematic regions during floral transition. We discuss this issue again below in the context of the specific points raised by the reviewer.

What are the responsible cells? Does the SAM accumulate more stem cells, or is it rather additional cell proliferation at the periphery? Which cells respond to the AP2/SOC1 instruction? Does regionally restricted expression of the pair phenocopy the doming? Can you extend the time of doming by ectopic expression? What are the signaling mechanisms downstream of AP2 and SOC1, for example is cytokinin involved? Since many of the mechanisms controlling cell behavior in the meristem are known, I would very strongly encourage the authors to include some data on these aspects.

The reviewer encourages us to include more analysis of the effects of AP2 and SOC1 on the cellular aspects of meristem development. We have approached these issues by first analyzing a *CLV3* transcriptional marker that defines the central zone and a *WUS* marker that defines the organising centre through the floral transition of Col-0 and scoring the changes in the spatial expression in detail. In addition, we built a computational pipeline to study *CLV3* and *WUS* expression at the tissue level during floral transition. This analysis demonstrated that in wild-type Col-0 the *CLV3* region extends in size, both in height and width, during floral transition. By contrast, *WUS* expression domain stretches vertically and transiently during the transition and maintains its width. We then introgressed the *CLV3* and *WUS* markers into *ap2* and *rAP2* genotypes. Using this material, we assessed how the dynamic changes in the *CLV3* and *WUS* domains of expression are affected by altering AP2 activity. We found that in the *ap2* mutant, the largest differences are

found in the early inflorescence meristem in the height and width of the *CLV3* domain and in the width of the peripheral zone. Whereas *rAP2-V* mainly causes an increase in width of the *CLV3* and *WUS* domains as well as the peripheral zone. We have added an entirely new Figure 3 and a new Supplementary Figure 6 to present these data, and added a new section to the Results on pages 8-10 describing the data in detail. Also, we added new panels in Supplementary Figure 11 and the new Methods section “Fluorescence quantification of *WUS::3xVENUS-NLS* and *CLV3::mCHERRY-NLS* “ to explain the newly developed pipeline.

Minor point: Please include top views of the AP2 and SOC1 reporters to allow the reader assess their 3D expression patterns.

We have included top views of AP2 in Figure 6 along with the time courses of the longitudinal view.

The top view of SOC1 reporter has been included in Figure 4 along with the longitudinal view.

Reviewer #2 (Remarks to the Author):

This manuscript dissects the role of AP2 in shoot doming during the developmental transitions that lead from the vegetative to the reproductive phase in Arabidopsis.

The first part of the paper focusses on the role of AP2 with section headings stating (1) AP2 positively regulates meristem area, (2) AP2 increases SAM height and width during the floral transition and (3) AP2 protein is present in the SAM during the floral transition.

Major concern for this section.

A. There is strong overlap between the data here and prior papers from the Coupland lab, primarily Sang et al 2022 (<https://doi.org/10.1111/nph.18111>) and for one aspect (AP2-VENUS accumulation during the floral transition) Ó'Maoiléidigh et al 2021(<https://journals.plos.org/plosbiology/article?id=10.1371/journal.pbio.3001043>).

We understand the reviewer’s point that previous work has described AP2 as a regulator of SAM size and its expression pattern in the SAM, but we believe that the results and conclusions of this manuscript differ substantially from previous publications on AP2 in the meristem, including the papers mentioned by the reviewer, and we have tried to bring this out more clearly in the revised manuscript. The current manuscript focuses on the role of AP2 in regulating SAM shape during floral transition. We emphasize that it is during floral transition that the major effect of AP2 on meristem height and width occurs. This is at first not intuitive, because AP2 is a repressor of flowering and it was not initially expected to have an important role during floral transition. The paper of Sang et al 2022 mentioned by the reviewer focuses on a much later stage of reproduction after bolting and inflorescence development when AP2 is already absent from the meristem and considers only meristem size. Therefore, that paper does not attempt to relate the precise time of

AP2 expression and action to the changes in meristem morphology during floral transition. The paper of Ó'Maoiléidigh et al 2021 focuses on the role of miR172 in repressing AP2 expression, and the temporal pattern of AP2 is not described at the same temporal resolution during floral transition as in the current manuscript, and is not quantified, as described in more detail below. These two papers are cited in our manuscript and we have made sure that they are cited at the appropriate places.

In particular Figures 1-3 in Sang et al correspond to data in Figures 1 and 2 in the current paper, except that the effect of AP1:VENUS and rAP2:VENUS accumulation was assayed during the floral transition in long days in Fig. 1 in Sang et al (day 12, day 15 (first flower)).

Figure 1 of Sang shows expression of AP2-V at one time point before floral transition (12LD) and its absence after floral transition (15LD), and discusses the anti-correlation with miR172, but does not examine the relationship between floral transition, meristem morphology and AP2 expression. No quantification of the dynamics of reduction in AP2-V is shown. We believe that to understand the effect of AP2 on SAM morphology during floral transition, it is important to show the quantification of the reduction in AP2-V expression, and to analyze it at higher temporal resolution than was done previously. Therefore, we have retained those data in Figure 2. However, we accept that showing AP2-V expression after transfer from SDs to LDs in the main figures is repetitive, and we now show those data and quantification of that time course in the Supplementary Figures (Supplementary Figure 3). We would like to show it there because it allows us to precisely time the reduction in AP2 abundance relative to the initiation of the flowering process by exposure to LDs. We discuss these data on pages 6-7.

Figure 2 of Sang describes flowering time and flower number of *ap2* and the complementing transgenic lines as well as *rAP2* plants. These data and materials are used as a basis for the present work but the data are not replicated. The flowering time of *soc1 ap2* is discussed below.

Figure 3 of Sang shows the meristem size of various genotypes at the mature inflorescence stage (1cm bolting). This is a much later stage than we analyze in the current paper, where we focus on the dynamic effects of AP2 on meristem shape during floral transition. Also, the main message of Figure 3 of Sang is the effect of miR172 on inflorescence meristem size, whereas the current paper focuses on meristem shape defined by height and width.

In summary, the current paper focuses specifically on the role and regulation of AP2 in conferring changes in meristem shape during floral transition, whereas Sang et al focused on meristem size during the later inflorescence stage and not on shape. Nevertheless, we have reduced further the perceived overlap by removing the AP2-V and rAP2-V expression data in the photoperiod shift experiment from the main figures and including an entirely new Figure 3 on quantification of the *CLV3* and *WUS* expression domains in Col-0, *ap2-12* and *rAP2-V* during floral transition.

In addition, Fig. 5 in Ó'Maoiléidigh showed that AP2-VENUS was still present in the SAM at day 14 in long days, that is to say during/after the floral transition. The authors use florally non-inductive short days plus photoperiod treatment here in a time course and come to the same conclusion (Fig. 2, heading 3 above).

As discussed above, we think it is important to show AP2-V expression in a high-resolution time course and to quantify the data in order to correlate AP2 expression with the changes in meristem shape, and have retained this in Figure 2. This is very different to the experiment of Fig. 5 in

Ó'Maoiléidigh, where, as the reviewer points out, “during/after the floral transition” could not be distinguished. The point of the Ó'Maoiléidigh figure is to illustrate the anti-correlation with miR172 expression, which is not demonstrated here.

In Fig. 3, Sang et al. measured the meristem size, cell number and cell area in wild type, *ap2* *rAP2:VENUS* plants in 1 cm bolted plants grown in long days, concluding that the meristem size and cell number are reduced in *ap2* and increased in *rAP2-VENUS* relative to the wild type, while the cell area in the L1 area is increased in both *ap2* and *rAP2-VENUS*. The same conclusions are drawn in Fig. 1 of the current manuscript for short day grown plants (heading 1 and 2 above).

The analysis of Sang was done in plants after floral transition in the inflorescence meristem at 1 cm bolting and focused on meristem size/area. We have reorganised Figure 1 to make it clearer that we are emphasizing meristem shape in width and height and that area is shown for comparison with the shape data. In addition, we score these changes during the floral transition and not at the much later stage reported by Sang. The conclusion is not the same as it addresses meristem shape during floral transition, and we have edited the heading, the text and reorganised the figure to make that clearer.

In Fig. 2 Sang et al. looked at changes in developmental timing in Col, *ap2*, *AP2:VENUS*, *AP2:VENUS ap2*, *rAP2:VENUS*, and *rAP2:VENUS ap2* (rosette leaf number, rosette and cauline leaf number (= total leaf number), which is also reported in Fig. 1d in the current manuscript. Thus, major conclusions ascribed to the data in the current paper had already been reached in prior publications by the same lab.

Figure 1d (now 1c) in the current paper focuses on the shape of the SAM at different stages of the floral transition and shows the identity of organ primordia present at each stage. It conveys different information than Figure 2 of Sang, which focuses on the number of mature organs present on the mature shoot of different genotypes after flowering.

B. The authors report that changes in meristem shape, in particular doming (vertical increases) are coupled to the floral transition by AP2 and SOC1 (see below). However, Fig. 1 d and Fig. S5 show clearly that the highest doming is first achieved around day 7 in all three genotypes tested. At this stage, the wild type has made many cauline leaves (end of ‘transition stage’), *ap2* is making flowers (is now in what the authors call the ‘inflorescence stage’), while *rAP2-VENUS* is just starting to exit vegetative development. At d7 all genotypes have also reached maximum width except the *rAP2* gain-of-function mutant, of all genotypes this one tested uniquely decouples doming (height increase) from width increase (Figure S4 and S6, current manuscript). The combined data argue strongly that doming and meristem size are not linked to developmental fate transition, they are decoupled – this contradicts the main conclusion and title of the paper.

We used the term coupling to describe the situation in wild-type Arabidopsis, where the increase in meristem height occurs during floral transition. As the reviewer points out, in the mutants these processes become uncoupled, emphasizing that AP2 has a role in coupling them. In particular, we argue that in *ap2* mutants the large increase in meristem height observed in Col-0 does not occur, so that AP2 is required for the extreme increase in height observed in Col-0. The *rAP2* plants express AP2 at elevated levels and their meristem increases in height as in Col-0, but the taller

meristem is retained for longer, and it becomes wider, so removing AP2 after floral transition is required to restrict doming to the floral transition. A similar effect is observed in *soc1* mutants, where AP2 mRNA persists for longer. We believe that the use of coupling in the context of Col-0 is valid, however we have removed the term from the title and emphasized it less in the text.

Part two headings state (4) AP2 delays the floral transition by repressing SOC1 before floral transition and (5) The interaction between SOC1 and AP2 coupled changes SAM morphology to the floral transition.

Major concerns:

C. The authors identify SOC1 using genomic approaches as a candidate AP2 target based on elevated expression in *ap2* mutants and during the floral transition and the locus being bound by AP2. All these findings were known about SOC1 from prior genomic datasets (Matthieu 2009, Yant 2009, Schmidt 2003). Yant had also shown genetically that SOC1 acts downstream of AP2, but *soc1* is not fully epistatic to *ap2*.

Yant et al analyzed “inflorescence transcriptomes in *Arabidopsis thaliana ap2-6* and Col-0 plants” whereas we used a time course of *ap2-12* apices harvested “under continuous LDs for 10, 12, 14 and 17 LD”. We think this is an important distinction because it allows us to analyze the effect of AP2 directly at the time when *ap2* mutants and Col-0 diverge in meristem shape. We also go on to analyze SOC1-GFP expression quantitatively in the meristem during floral transition in wild-type and *ap2-12* mutant backgrounds providing much more spatial-temporal information on the regulation of SOC1 by AP2 than was previously available. Yant et al did describe the *soc1 ap2* double mutant, which we have cited, but only scored total leaf number. We find it important here to describe for this genotype all aspects of flowering, bolting time, rosette leaf number and cauline leaf number, as well as meristem shape during flowering. Nevertheless, we have moved the flowering time data to Supplementary Figure 8 to avoid an apparent overlap with the work of Yant et al in the main figures. Matthieu et al 2009 focus on SMZ and its regulation by miR172, so there does not seem to be an overlap. The paper of Schmid et al (2003) performed on microarrays did not analyze the *ap2* mutant, so there does not appear to us to be an overlap, although its impact on the transcriptomics of floral transition was mentioned. We have cited Schmid et al (2003) and Yant et al (2010).

Moreover, the Coupland lab had linked SOC1 to promotion of doming via direct positive regulation of gibberellin biosynthesis (Kinoshita et al, 2020 <https://elifesciences.org/articles/60661>).

Using the exact same photoperiod shift setup as in the current paper, *soc1* mutants and gibberellin biosynthesis mutants show reduced meristem area already at + 5 long day (Kinoshita et al), prior to the stage when AP2 is reduced (+7 long day, current manuscript). Likewise, SOC1-GFP levels increase in wild type around day 12 in long days (Fig. 3 current manuscript), with AP2-VENUS still unchanged (Fig. 5b current manuscript), these data do not support heading 4.

This comment relates to heading 4 in the previous version “AP2 delays floral transition by repressing SOC1 expression before floral transition”. The evidence for this is that in the RNAseq

and in the confocal microscopy *SOC1* expression is higher in an *ap2* mutant at early time points than in Col-0, and that inactivating *SOC1* in *soc1 ap2* double mutants delays the early flowering phenotype of *ap2* mutants. Nevertheless, this heading has been deleted in the new version and the whole analysis of interactions between *SOC1* and *AP2* rewritten into two sections “Mutual repression of *SOC1* and *AP2* contributes to the regulation of SAM morphology during floral transition” and “Mutual repression of *SOC1* and *AP2* affects flowering time and floral primordium identity”. In the latter section, we have included new genotypes of *rAP2-V soc1* to show the additive effects of *soc1* on the late flowering of *rAP2-V*.

Generally, concerning the cross regulation of *SOC1* and *AP2*, we find it most convincing to consider the genetic interactions. In the *ap2* mutant, *SOC1* expression is increased before floral transition (see Figures 4c-d and Supplementary Figure 7b) and this contributes to the early flowering (see Supplementary Figure 8e-h). Also, the larger meristem of *soc1* is partially dependent on increased *AP2* expression (see Figure 5a-c, Supplementary Figure 10 and Figure 6a-b), and increased *SOC1* in *ap2* mutants can reduce SAM width.

We agree that we should have mentioned the effect of *SOC1* on gibberellin biosynthesis and have now included this in the Introduction and the Discussion.

Finally in wild-type morphology plants (*AP2-VENUS* rescues *ap2*, Sang et al), *AP2* expression is fully downregulated only at day 17; Fig. 5b), while first flowers form in the wild type already at day 14 (Fig. 4b), inconsistent with the model proposed (Fig. 6).

The mean value of *AP2* expression is already reduced at day 14. Most commonly, the first flowers are detected in Col-0 at day 17, see for example Figure 1c, when *AP2* is completely downregulated, and at day 14 cauline leaves are still being formed, see Figure 1c. There is variation in the flowering stage of individual plants at each time point, and to make the stages clearer we have now included the number of plants at each stage in Figure 1c. We do not find this inconsistent with the model, and believe the inclusion of the data on the number of plants at each stage makes this clearer.

D. As mentioned above in B for photoperiod shifts the change in meristem size and doming in are decoupled from the developmental transitions that lead to flower formation. These data call in question whether the interaction between *SOC1* and *AP2* couples changes SAM morphology to the floral transition (heading 5).

We consider these processes coupled in Col-0, but uncoupled in *ap2* mutant which flowers early but does not form a strongly domed meristem. Nevertheless, we have removed the term “couples” from the title and most of the text. As described above, we have altered the main title and the titles of the sections on the interactions between *AP2* and *SOC1* to remove this term. We have also more thoroughly described the interpretation of the meristem shape changes of *ap2-12 soc1-2* compared to *soc1-2* and *ap2-12* during floral transition in Figure 5 and on pages 12-13 to make clearer their interaction.

Additional comments

1. For Fig 2, the authors should mention the number of samples they examined on each day. Or they could perform image quantification as they did for SOC1-GFP.

In the previous version, Figure 2 was AP2-VENUS expression after transfer of plants from SD to LD. As mentioned above, we removed this analysis of from the main figures. Another example of such a time course now appears in Supplementary Figure 3, and we show quantification of that as requested, together with the corresponding sample size.

2. For Fig 3, SOC1-GFP in wild type background shows weaker expression compared to that in *ap2* background on day 10 and day 12, which does not match the phenotypic data where *ap2* has smaller meristem than WT from day 14 onwards (Supplementary Fig 5b). It seems SOC1 expression is released in *ap2* mutant only before day 14, while the phenotypic change in *ap2* mutant happens only after day 14.

Yes, SOC1 expression is increased earlier in *ap2* mutant, and this likely contributes to the early flowering because the *soc1-2 ap2-12* double mutant flowering is delayed. However, the increased SOC1 expression is not sufficient to reduce meristem size in *ap2* during vegetative development, this requires other factors that are expressed during floral transition. However, as shown in the quantified data now included in Figure 5b-c, the *soc1-2 ap2-12* double mutant shows a taller and wider meristem than *ap2-12* at day 14, suggesting that active SOC1 alters meristem shape in *ap2-12* at this time point. SOC1 is expressed at this time point in *ap2-12*, probably at a higher level than Col-0 although it is not significantly higher in the statistical analysis, but the phenotypic change could also be a consequence of the statistically higher expression at the earlier time point of 12 LD. We have clarified the text on this point in the section "Mutual repression of SOC1 and AP2 contributes to the regulation of SAM morphology during floral transition".

3. To thoroughly test the mutual regulation between SOC1 and AP2, the SOC1 binding sites on AP2-Venus and mutating the AP2 binding sites on SOC1-GFP should be mutated. This excludes indirect effects the two known regulators of the floral transition and would be novel information.

This experiment is more complicated than it seems and could not be done in the time of a revision. Identifying the precise binding sites under the ChIP peaks has not been done, and there might be redundancy for them. Moreover, binding sites for MADS box transcription factors and probably for AP2-LIKE factors are used repeatedly by different proteins in the same class, which can make the phenotypes of such mutants difficult to interpret. Importantly, to generate transgenic plants with such mutations and to quantitatively analyse them could easily take a year if not longer, and the result would be uncertain for the reasons mentioned. We consider this experiment therefore to be outside the scope of this revision. We have included other novel information in an entirely new Figure 3 and by incorporating newly described genotypes in Fig. 5d-e and Supplementary Figure 9.

4. Since there are differences in the duration of developmental stages in different lines, comparing the same fluorescent reporters in different backgrounds on the same day might just be comparing fluorescence reporters at different stages of plant development (e.g. Fig 2, 3d, 5a). This could be avoided using transient gain or loss of function.

The reviewer is right that in the analysis of flowering time mutants the timing of developmental stages can also be changed. However, we have included detailed analysis of meristem shapes through time courses and the timing of cauline leaf and flower development related to shape in Figures 1c and Figures 5c, d. We have also included a new Figure 2d in which we compare the meristem shape and flowering stage of each genotype at the same developmental stage, the time point at which maximum meristem height was attained. In this way, the meristems of individual plants and their flowering stage can be compared across genotypes at a common developmental stage defined by meristem shape. Moreover, the inclusion in Figure 2d and Supplementary Figure 5 of the *syp* mutant that flowers early but still shows a highly domed meristem addresses the issue of whether early flowering causes the reduced meristem height of the *ap2* mutant. We have described these data in the section “Relationship of organ primordia identity to meristem shape during floral transition” on pages 7-8. Transient induction experiments might eventually help resolve some of these issues, but would take too long to include in the scope of this revision.

5. For the image quantification method, the authors mentioned “the paraboloid curvature was increased with respect to the original such as $a'=a/\alpha$ ”. The author should show the exact value of α and how they determine the value. And it is better to show an example of quantified region in a single confocal slice in Supplementary Fig 11 (e.g. masking the final region of quantification in Supplementary Fig 11d).

We have edited the explanation of the use of the parabola curvatures to build the 3D-paraboloid in Methods, in the section “Fluorescence quantification of SOC1-GFP and AP2-VENUS” so that all the magnitudes used in this conversion process are clear. The Greek letter α accounts for the resolution of the image. We agree that in the previous version this was not clear and we have edited the text to read:

“To exclude any fluorescence signal at the boundaries of the SAM and primordia, the paraboloid curvature was increased with respect to the original such as $a'=a/\alpha$, where a' is the curvature of the new paraboloid and α the image resolution ($\alpha < 1 \mu\text{m}$).”

As requested, we have also included in Supplementary Figure 12 a single confocal slice masking the quantified region.

Reviewer #3 (Remarks to the Author):

During the floral transition, SAM enlarges and gives rise to floral primordia instead of leaf primordia; however, how the meristem size and shape changes during the floral transition are regulated remains largely unknown. The manuscript is entitled, “A regulatory gene network that couples floral transition to shoot apical meristem morphology in Arabidopsis”. The authors found that SOC1 is activated during floral transition, and then represses the expression of AP2. AP2 is an essential regulator for meristem enlarge during the transition stage. The high level and the duration of AP2 expression enable a larger size of SAM. The authors establish a SOC1-AP2 feedback regulatory network to ensure the proper size of SAM during the floral transition. I have some questions and suggestions as mentioned below.

1. The size of the meristem greatly changes during the floral transition but the change mostly

disappears in the *ap2* mutant, indicating AP2 is an essential regulator for meristem enlarge during transition. As we know, the plants flowering early are more likely with a small meristem, and vice versa. The different meristem sizes with altered flowering times are all due to the changes in the expression level of AP2? What is the AP2 expression in other flowering-time mutants?

The reviewer asks whether AP2 regulation contributes to meristem size in other flowering time mutants. This is difficult to study using AP2 mRNA level because AP2 is regulated by miR172 at the post-transcriptional level as well as at the transcriptional level. So, it would be necessary to cross the AP2-V marker into flowering time mutants to test this. We present these data in detail for *soc1*, and find that AP2 expression is required for the increased meristem height and width of *soc1* mutants, but that *ap2 soc1* mutants still show increased width in comparison to *ap2*. So, even in *soc1* mutants AP2 does not explain all of the effect on meristem shape. To ask whether all early flowering mutants have a small meristem that like *ap2* does not increase in height and width to the same extent as Col-o, we tested the *svp* mutant, which flowers around the same time as *ap2*. We found that the meristem of *svp* reaches a similar height to that of Col-0, and therefore the reduced doming in *ap2* mutants is not only a feature of its early flowering. We show these data in in Supplementary Figure 5 and describe them on page 8.

2. The flowering time of WT, *ap2-12*, *soc1-2*, and *ap2-12 soc1-2* are all different. The meristem size changes a lot during the floral transition. So, I suggest the authors give a guideline on how to compare the meristem size of plants with different flowering time.

We understand that this is a complex issue because the different flowering times might also affect meristem size changes in the genotypes. Therefore, we have included a new figure that aims to compare meristems at the same stage for each genotype. We selected the stage at which the meristem of the genotype reaches its maximum height, because that is a recognisable stage during floral transition for each genotype. We then represented the height and width as well as the developmental identity of the primordia present (leaf, cauline leaf, floral primordia). We have included this figure as Fig. 2d, and mention it in the Results and Discussion, for example on pages 7-8.

The meristem size of *ap2-12 soc1-2* is rescued. Is it because of the partially rescued flowering time?

The reviewer is right that *ap2-12 soc1-2* shows a meristem area intermediate between *soc1-2* and *ap2-12* and close to Col-0 at some time points (now Supplementary Figure 8b). This issue is also evident in the newly incorporated data on meristem height and width in *soc1-2 ap2-12* (Figure 5b, c) where it is also true that the maximum height and width of *soc1-2* meristems are reduced in *soc1-2 ap2-12*, but at most time points are still larger than *ap2-12*. The intermediate flowering time might contribute to this, but as we discuss on page 8, the comparison with the *svp* mutant suggests that AP2 has two separable functions, one to repress flowering and another to promote meristem size, and we discuss the genetic interaction with *soc1* in that context, for example on pages 12-13. Similarly, although the flowering time of *soc1-2 ap2-12* is later than Col-0 in terms of rosette leaf number and bolting time in Supplementary Figure 8e-g, particularly meristem height at maximum height is smaller in Figure 2d.

3. The authors mention two growth conditions, one is “grown for 2 weeks under non-inductive short

days (2 wSD) then exposure to LDs”, and the other is “grown under continuous LD”. I hope the authors will explain the reasons for setting these two conditions in detail.

The reviewer is right that the original version was confusing moving between these two conditions. In the revised version, we have focused all of the main figures on plants grown under long days continuously from germination. We have included the phenotypic analysis of Col-0, *ap2-12* and *rAP2-V* after transfer from short days to long days in Supplementary Figure 2, and the effect of transfer from short days to long days on *AP2-V* and *rAP2-V* expression in Supplementary Figure 3. The transfer between photoperiod conditions allows floral transition to be staged from transfer rather than from germination. This also enables comparison of phenotypes with previously published results such as those of Kinoshita et al 2020. However, as continuous LDs was used for the transcriptome experiment and the flowering time data, it is more appropriate to use continuous LDs in all of the main figures.

4. The authors predict AP2 regulates meristem size during the floral transition most through increasing *WUS* transcription. Do authors examine the *WUS* expression in WT and AP2 during the transition stage?

We have now included a full analysis of a *WUS* marker and a *CLV3* marker during floral transition in Col-0, *ap2* and *rAP2* as new Figure 3 and Supplementary Figure 6. This analysis allows us to determine the meristematic regions that change during floral transition in Col-0 and to compare these with *ap2* and *rAP2*. We provide a full description of these data on pages 8-10 and discuss the relationship between *AP2* and *WUS* in detail on pages 14-15 emphasizing that this is likely to be complex and that *AP2* does not regulate meristem shape by simply activating *WUS*.

5. What is the meristem phenotype of 35S:*rAP2*? It can rescue the meristem size of 35S:*SOC1*?

We have now included a description of the meristem of 35S:*SOC1* in Figure 5d, Figure 2d and Supplementary Figure 9a, b. These data show the meristem of 35S:*SOC1* is reduced in width and height compared to Col-0, as expected. We have not analyzed 35S:*rAP2* because these plants show strongly reduced fertility and altered development. However, we have included data for *rAP2 soc1-2* in Figure 2d, Figure 5e and Supplementary Figure 9c-g. This genotype shows a higher and wider meristem than both progenitors, consistent with increasing *AP2* expression both by reducing inhibition by *miR172* and by reducing transcriptional repression by *SOC1*.

Minor points

1. Please explain why setting p value to 0.075 rather than 0.05 in Figure 5.

We have now established 0.05 as the threshold of p-values for all the comparisons.

2. In Figures 1a,b, c, Why the *ap2-12* data in +11LD is missing?

The *ap2-12* mutants flowered earlier, and therefore by +11LD they were already well advanced in inflorescence development. We did not consider that this time point added

information on floral transition. These data have been moved to Supplementary Figure 2 in the revised version, and we have included this point in the legend.

REVIEWER COMMENTS

Reviewer #1 (Remarks to the Author):

The authors have included an important set of new data revealing the behavior of the stem cell regulators WUS and CLV3 during the transition and have thus fully addressed my concerns.

I support publication of manuscript in its current form.

Reviewer #2 (Remarks to the Author):

Some of the findings reported here have been published already. This includes repression of AP2 and SOC1 and vice versa, and their opposite roles in the developmental transitions that lead to flower formation. This matters here as the manuscript as a whole is rather descriptive.

Some of the shoot measurements the authors present are interesting and novel. And there are obvious defects in doming in *ap2*. Yet, I struggle with general conclusions about links between doming and the timing of developmental transitions (coordination of both processes as indicated in the title). *ap2* mutants do not really dome and form flowers more rapidly; the latter is expected as AP2 is a repressor of the floral transition. *rAP2-V* and *soc1* still dome, like wild type, and all form flowers after doming. Is there a downstream defect on developmental transition caused by the loss of doming in *ap2* mutants? In other words what is the function of doming?

Cause and consequence are also tricky for the CZ, OC and PZ. The CZ and SAM size change in the same manner, a ratio between these would likely be close to invariant. The OC does not change over time but is surrounded by a wider PZ. How are these changes coupled, does one drive the other (which?). How can we say AP2 regulates the CZ size for example, it could also be the case that altered SAM size leads to a change in the CZ.

Related to the relative timing of events. The authors throughout compare age matched plants and not developmental stage matched plants, the latter would be based on developmental hallmarks such as the floral transition, first flower formed etc. Age matching (same day after germination) means that many of the phenomena reported are expected to be delayed/precocious in mutants that transition later/earlier to flower fate., this should be considered in the interpretation of the data.

Reviewer #3 (Remarks to the Author):

In this extensively revised manuscript, the authors present evidence elucidating the interplay between AP2 and SOC1 in regulating meristem morphology during the floral transition. They demonstrate significant enlargement in both width and height of the meristem during this crucial period. By incorporating WUS and CLV3 markers, they effectively depict the dynamic changes occurring in specific regions of SAM. Furthermore, the inclusion of an early flowering mutant, *svp*, helps to underscore the specific role of AP2 in meristem size regulation. Overall, the manuscript has undergone substantial improvements.

1. In multiple experiments, the meristem size in the Col shows a gradual increase from 10 to 14 LD, followed by a noticeable decrease. However, in the *ap2* mutant, this initial increase does not happen, and lateral-stage shrinkage of SAM is also absent. Surprisingly, the final meristem size of both WT and *ap2* mutants is slightly different. Additionally, the expression level of AP2 consistently decreases from 14LD onwards. So, the question arises: Does AP2 primarily promote meristem size or maintain it?
2. Given the described phenotype, the notable disparity in SAM size between WT and *ap2* occurs within a relatively short window from 12LD to 19LD. How might this discrepancy impact lateral plant development?
3. There are some format errors in the discussion part (page 16)

Response to the Reviewers

We would like to thank the reviewers for considering the revised version of the manuscript, and for their constructive comments on the text and data. We respond to each of their comments below. Our responses are in bold and the reviewers' comments appear verbatim.

REVIEWER COMMENTS

Reviewer #1 (Remarks to the Author):

The authors have included an important set of new data revealing the behavior of the stem cell regulators WUS and CLV3 during the transition and have thus fully addressed my concerns. I support publication of manuscript in its current form.

We are pleased that reviewer #1 found the new data set important and that they recommended publication of the manuscript.

Reviewer #2 (Remarks to the Author):

Some of the findings reported here have been published already. This includes repression of AP2 and SOC1 and vice versa, and their opposite roles in the developmental transitions that lead to flower formation. This matters here as the manuscript as a whole is rather descriptive.

We understand that the reviewer finds it important to stress this point, which was mentioned also in their first review. The previous data on the interactions between SOC1 and AP2 are all cited and acknowledged. We would like to stress that our data builds on these previous results, but also goes much further. The interactions between SOC1 and AP2 on meristem shape and size is analyzed here in the single and double mutants, and the meristem of the double mutant had not previously been analyzed. Moreover, we score the flowering phenotype in much more detail, including bolting time, rosette leaf number and cauline leaf number measurements. Finally, the expression patterns of each gene in the background of mutation of the other gene is analyzed quantitatively at the protein level in the shoot meristem through a time course of floral transition. Previous analyses were based on transcriptomic data or RT-qPCR, so that our data describes the mutual regulation of these genes at much higher spatiotemporal resolution and at a different regulatory level than was done previously. We consider that our work greatly extends the earlier interesting observations and places them in the context of the SAM.

Some of the shoot measurements the authors present are interesting and novel. And there are obvious defects in doming in *ap2*. Yet, I struggle with general conclusions about links between doming and the timing of developmental transitions (coordination of both processes as indicated in the title). *ap2* mutants do not really dome and form flowers more rapidly; the latter is expected as AP2 is a repressor of the floral transition. *rAP2-V* and *soc1* still dome, like wild type, and all form flowers after doming. Is there a downstream defect on developmental transition caused by the loss of doming in *ap2* mutants? In other words what is the function of doming?

We appreciate that the reviewer recognizes the novelty and interest in some of the shoot meristem measurements, and the accuracy of the description that *ap2* mutants are severely impaired in meristem shape. The coordination between increases in meristem height (doming) and

flowering clearly takes place in Columbia wild-type where the two processes are closely coupled and meristem doming is one of the first phenotypic changes observed during floral transition. The adaptive significance of doming is more difficult to address at this stage, and the reviewer is correct to point out that although the meristem of *ap2* mutants does not greatly increase in height, the mutant does undergo floral induction. Therefore, an increase in meristem height is not required for floral transition. Nevertheless, the associated increase in SAM size likely makes an important contribution to inflorescence development and to the number of flowers formed. Previously, *rAP2* plants were shown to form more flowers in the inflorescence at a faster rate than Col-0, and *ap2* mutants formed flowers at a slightly slower rate, but the effect of *ap2* on flower number is difficult to assess because of its secondary effect on flower fertility. The precise role of meristem doming on inflorescence development is outside of the scope of this manuscript, but there is a sentence in the Introduction that addresses this:

“Notably, during the transition to flowering, the SAM enlarges and changes identity to initiate the formation of flowers instead of leaves⁵⁻⁷. The increase in SAM size during floral transition persists in the inflorescence meristem, and likely contributes to the number of flowers formed^{5,8-10}.”

Cause and consequence are also tricky for the CZ, OC and PZ. The CZ and SAM size change in the same manner, a ratio between these would likely be close to invariant. The OC does not change over time but is surrounded by a wider PZ. How are these changes coupled, does one drive the other (which?). How can we say AP2 regulates the CZ size for example, it could also be the case that altered SAM size leads to a change in the CZ.

We agree with the reviewer that we cannot at present comment on cause and consequence, but the experiments we show are important in determining which meristematic regions change during floral transition and how these changes differ in the *ap2* and *rAP2* genotypes. The *WUS* and *CLV3* transgenes are used as markers to assess the changes in meristematic zones during floral transition in Col-0, *ap2* and *rAP2*. We find the results interesting and important, and they clearly indicate that a large difference in the PZ width occurs during floral transition and persists into the inflorescence. It is possible that the changes we observe are not due to functional alterations in the *WUS-CLV3* feedback loop or in stem cell activity but in the proliferation of cells in the PZ and boundary regions. We believe that our manuscript will be important in highlighting these issues, but much more experimentation will be required by us and other groups to distinguish these possibilities. These future experiments necessarily go beyond the scope of the present paper.

Related to the relative timing of events. The authors throughout compare age matched plants and not developmental stage matched plants, the latter would be based on developmental hallmarks such as the floral transition, first flower formed etc. Age matching (same day after germination) means that many of the phenomena reported are expected to be delayed/precocious in mutants that transition later/earlier to flower fate., this should be considered in the interpretation of the data.

At our current understanding we do not have enough morphological staging information to routinely match plants by other criteria apart from time. However, in response to this point in the previous revision, we did include an entirely new figure (Fig. 2d) in which we compared different genotypes at the same morphological stage of maximum meristem height. And it is clear that at that stage there are significant differences in meristem shape and size among the genotypes, fully supporting the role of AP2 in regulating meristem shape in addition to its role in repressing floral transition.

Reviewer #3 (Remarks to the Author):

In this extensively revised manuscript, the authors present evidence elucidating the interplay between AP2 and SOC1 in regulating meristem morphology during the floral transition. They demonstrate significant enlargement in both width and height of the meristem during this crucial period. By incorporating WUS and CLV3 markers, they effectively depict the dynamic changes occurring in specific regions of SAM. Furthermore, the inclusion of an early flowering mutant, *svp*, helps to underscore the specific role of AP2 in meristem size regulation. Overall, the manuscript has undergone substantial improvements.

We are pleased that the reviewer appreciated the new data sets and considered the manuscript substantially improved.

1. In multiple experiments, the meristem size in the Col shows a gradual increase from 10 to 14 LD, followed by a noticeable decrease. However, in the *ap2* mutant, this initial increase does not happen, and lateral-stage shrinkage of SAM is also absent. Surprisingly, the final meristem size of both WT and *ap2* mutants is slightly different. Additionally, the expression level of AP2 consistently decreases from 14LD onwards. So, the question arises: Does AP2 primarily promote meristem size or maintain it?

We propose that AP2 is a positive regulator of meristem size considering the reduction in meristem size and alterations in meristem shape in *ap2* mutants during the early stages of floral transition. However, the reviewer is right that AP2 can also be considered to enable a maintenance of meristem size, because persistence of AP2 expression in *rAP2* plants maintains the SAM at a larger size for longer. Nevertheless, we believe that based on the mutant phenotype of *ap2* mutants, the primary effect of AP2 in wild-type plants is to promote an increase in meristem height and width.

2. Given the described phenotype, the notable disparity in SAM size between WT and *ap2* occurs within a relatively short window from 12LD to 19LD. How might this discrepancy impact lateral plant development?

This point is similar to the one raised by reviewer 2 on the significance of meristem doming. Here, we would point to the effects on flower production and the sentences in the Introduction (“Notably, during the transition to flowering, the SAM enlarges and changes identity to initiate the formation of flowers instead of leaves⁵⁻⁷. The increase in SAM size during floral transition persists in the inflorescence meristem, and likely contributes to the number of flowers formed^{5,8-10}.”).

3. There are some format errors in the discussion part (page 16)

Thanks – we have corrected the errors and inserted the figure numbers.

REVIEWERS' COMMENTS

Reviewer #2 (Remarks to the Author):

I have no more comments for the authors. I do think the MS title needs revision, as the authors concur that it is "correct to point out that although the meristem of ap2 mutants does not greatly increase in height, the mutant does undergo floral induction. Therefore, an increase in meristem height is not required for floral transition."

Reviewer #3 (Remarks to the Author):

This manuscript has been well and satisfactorily revised, demonstrating the role of AP2 in coordinating meristem shape and identity. I believe it is ready for publication in its current form.

REVIEWERS' COMMENTS

We thank the reviewers for reviewing the revised versions of our manuscript. We respond to their comments below. Our responses are in bold and the reviewers' comments appear verbatim.

Reviewer #2 (Remarks to the Author):

I have no more comments for the authors. I do think the MS title needs revision, as the authors concur that it is "correct to point out that although the meristem of *ap2* mutants does not greatly increase in height, the mutant does undergo floral induction. Therefore, an increase in meristem height is not required for floral transition."

We carefully considered the reviewer's suggestion to change the title of the manuscript, but decided that in our opinion the current title accurately describes the content of the paper. In wild-type meristems doming and flowering time are closely coordinated, whereas in *ap2* mutants doming is strongly impaired and floral transition takes place earlier, consistent with the comment highlighted by the reviewer that "an increase in meristem height is not required for floral transition". Nevertheless, it seems reasonable to mention in the title that AP2 coordinates meristem shape and identity during floral transition, because this co-ordination is disrupted in *ap2* mutants compared to wild-type plants. Therefore, the title summarises the wild-type situation, and we do not think that "Coordination of shoot apical meristem shape and identity by APETALA2 during floral transition in Arabidopsis" is in conflict with the comment highlighted by the reviewer concerning the *ap2* mutant.

Reviewer #3 (Remarks to the Author):

This manuscript has been well and satisfactorily revised, demonstrating the role of AP2 in coordinating meristem shape and identity. I believe it is ready for publication in its current form.

We are glad that the reviewer acknowledges the revision of the manuscript and recommends the publication of the manuscript in its current form.